# FAST: a Fused and Accurate Shrinkage Tree for Heterogeneous Treatment Effects Estimation

Jia Gu[*1,2], Caizhi Tang[*2], Han Yan[3], Qing Cui[2], Longfei Li[2], and Jun Zhou[†2]

[1]Center for Statistical Science, Peking University
[2]Ant Group
[3]Guanghua School of Management, Peking University
gujia@pku.edu.cn, caizhi.tcz@antgroup.com, hanyan@stu.pku.edu.cn, {cuiqing.cq, longyao.llf, jun.zhoujun}@antgroup.com

## Abstract

This paper proposes a novel strategy for estimating the heterogeneous treatment effect called the Fused and Accurate Shrinkage Tree (FAST). Our approach utilizes both trial and observational data to improve the accuracy and robustness of the estimator. Inspired by the concept of shrinkage estimation in statistics, we develop an optimal weighting scheme and a corresponding estimator that balances the unbiased estimator based on the trial data with the potentially biased estimator based on the observational data. Specifically, combined with tree-based techniques, we introduce a new split criterion that utilizes both trial data and observational data to more accurately estimate the treatment effect. Furthermore, we confirm the consistency of our proposed tree-based estimator and demonstrate the effectiveness of our criterion in reducing prediction error through theoretical analysis. The advantageous finite sample performance of the FAST and its ensemble version over existing methods is demonstrated via simulations and real data analysis.

## 1 Introduction

Causal effects are the magnitude of the response of an effect variable (also called outcome) caused by the effect variable (also called treatment), which is a fundamental and essential issue in the field of casual inference (Imbens and Rubin, 2016). And the heterogeneous treatment effect (abbr. HTE) is usually used to characterize the heterogeneity of causal effects across different subgroups of the population. In recent years, heterogeneous treatment effect estimation has been successfully applied in various fields such as epidemiology, medicine, and social sciences (Glass et al., 2013; Kosorok and Laber, 2019; Turney and Wildeman, 2015; Taddy et al., 2016).

In general, the causal problems can be studied through both experimental studies (also known as randomized control trials, RCTs) and observational studies. Experimental studies are widely regarded as the gold standard for assessing causal effects since the randomization process eliminates the possibility of confounding bias. However, large-scale RCTs can be challenging due to issues related to cost, time, and ethics (Edwards et al., 1999). On the other hand, observational data are often readily available with an adequate sample size. Under certain fairly strong assumptions, such as unconfoundedness assumption, there is a rich literature regarding the estimation of HTE in observational studies, such as tree-based methods (Athey and Imbens, 2016; Wager and Athey, 2018; Athey et al., 2019; Hahn et al., 2020), boosting (Powers et al., 2017), meta learners (Künzel et al.,

---

[*]Equal Contribution
[†]Corresponding Author

37th Conference on Neural Information Processing Systems (NeurIPS 2023).

2019) and $R$-learner (Nie and Wager, 2020). However, the unconfoundedness assumption, which requires measuring all confounders, is in general untestable unless extra information such as the existing of instrumental variables is available (de Luna and Johansson, 2014). And any violations of this assumption may result in seriously invalid causal statements. Various methods have been proposed to mitigate the unmeasured confounding in observational studies, such as the sensitivity analysis (Rosenbaum and Rubin, 1983; Zhang and Tchetgen Tchetgen, 2022), the instrumental variables (IV) approach (Angrist et al., 1996) and the proximal causal inference (Kuroki and Pearl, 2014; Miao et al., 2018; Shi et al., 2020; Cui et al., 2023). However, the validity of these procedures also relies crucially on assumptions that are often difficult to verify in practice.

Given the limitations of relying on individual data sources, data fusion, as a branch of causal inference strategies that integrates both the trial and the observational data, has gained significant interest in the literature (Bareinboim and Pearl, 2016; Colnet et al., 2020; Shi et al., 2022). Existing data fusion methods for estimating the HTE include the KPS estimator obtained by modeling the confounding function parametrically (Kallus et al., 2018), the semi-parametric integrative estimator under the parametric structural models (Yang et al., 2020) and the integrative R-learner (Wu and Yang, 2022). Besides, (Tang et al., 2022) proposed the Gradient Boosting Causal Tree (GBCT), which integrated the current observational data and their historical controls for estimating the conditional average treatment effect on the treated group (CATT).

This paper presents a novel approach for estimating heterogeneous treatment effects (HTE) in the context of causal data fusion. The proposed method, named Fused and Accurate Shrinkage Tree (FAST), *avoids* the need for a two-stage estimation process required in conventional data fusion strategies, which involves modeling and estimating the nuisance confounding bias function. The main contributions of this work can be summarized as follows (i) The authors propose a novel shrinkage method for combining an unbiased and biased estimator, which effectively reduces the mean square error of the unbiased estimator, and provides an easy implementation of the method tailored for the HTE estimation; (ii) The authors extend the conventional node split criterion via a re-scaling technique, which automatically penalizes the use of the observational data with low quality (namely large confounding bias); (iii) The authors also provide a theoretical analysis to explain the advantages of our splitting criterion.

## 2 Background and motivation

### 2.1 Notations

Let $\boldsymbol{X} \in \mathcal{X} = [-1,1]^p$ be a $p$-dimensional vector of pre-treatment covariates, $\boldsymbol{U} \in \mathbb{R}^q$ ($q \geq 0$) be a possibly unmeasured random vector consisting of the confounding variables, $D$ be a binary treatment variable ($D = 0$ denotes the control and $D = 1$ denotes the treated) and let $Y(d)$ be the potential outcome that would be observed when the treatment had been set to $d \in \{0,1\}$. We follow the potential outcome framework (Rubin, 1974) to define the heterogeneous treatment effect $\tau(\boldsymbol{x}) = \mathbb{E}(Y(1) - Y(0)|\boldsymbol{X} = \boldsymbol{x})$.

Suppose that we can collect two kinds of data: trial data and observational data, and they are described by $n + m$ quadruples, $\{Y_i, D_i, \boldsymbol{X}_i, S_i\}_{i=1}^{n+m}$, where $S_i$ indicates if the $i$-th individual would have been recruited ($S = 1$) or not ($S = 0$) in the trial. We also denote $\mathcal{R} = \{1, 2, \cdots, n\}$ the set of indices of observations in the RCT study, and $\mathcal{O} = \{n+1, n+2, \cdots, n+m\}$ the set of indices of observations in the observational study. We define $e(\boldsymbol{X}, \boldsymbol{U}, S) = P(D = 1|\boldsymbol{X}, \boldsymbol{U}, S)$ as the propensity score of the trial and observational population, respectively. In practice, due to $\boldsymbol{U}$ being unknown, we usually use $\hat{e}(\boldsymbol{X}, S)$ to estimate $e(\boldsymbol{X}, \boldsymbol{U}, S)$. In addition, $\hat{e}(\boldsymbol{X}, 1)$ is unbiased, but $\hat{e}(\boldsymbol{X}, 0)$ is biased because the unmeasured confounder $\boldsymbol{U}$ in the observational study can be related to the assignment of treatment $D$. Let $\tau_1(\boldsymbol{x}) = \mathbb{E}(Y(1) - Y(0)|\boldsymbol{X} = \boldsymbol{x}, S = 1)$ be the HTE on the trial population. We then make the following fundamental assumption on the trial and observational studies, which facilitates the potential for causal data fusion:

**Assumption 1.** *(i) For any $\boldsymbol{x} \in \mathcal{X}$, $\tau_1(\boldsymbol{x}) = \tau(\boldsymbol{x})$; (ii) $Y(d) \perp D|(\boldsymbol{X}, S = 1)$ for $d \in \{0,1\}$ and (iii) the propensity score $\delta < e(\boldsymbol{X}, S) < 1 - \delta$ almost surely for some constant $0 < \delta < 1/2$.*

Assumption 1 (i) states that the HTE function is transportable from the trial population to the target population. Stronger versions of Assumption 1 include the ignorability of study participation (Buchanan et al., 2018) and the mean exchangeability (Dahabreh et al., 2019). In the following of this

paper, we use $|\Lambda|$ to denote the number of elements for any set $\Lambda$, $\lfloor c \rfloor$ to denote the biggest integer less than or equal to the constant $c$, and $[p]$ to denote the index set $\{1, 2, \cdots, \lfloor p \rfloor\}$. For two positive sequences $\{a_n\}_{n \geq 1}$ and $\{b_n\}_{n \geq 1}$, we write $a_n = \mathrm{O}(b_n)$ if $|a_n/b_n|$ is bounded.

## 2.2 Tree-based methods

To estimate the HTE, it is reasonable to perform subgroup analysis by appropriately stratifying or matching (Frangakis and Rubin, 2002) the samples into multiple subgroups that differ in the magnitude of treatment effects. In machine learning, tree-based methods (Breiman et al., 1984; Breiman, 2001; Friedman, 2001) are usually used for such stratification tasks, which greedily optimize the loss function, also called splitting criterion, via recursively partitioning feature space. In fact, many tree-based causal methods designed for the HTE estimation were also proposed (Radcliffe and Surry, 2012; Athey and Imbens, 2016; Athey et al., 2019). Along with the development of the tree-based methods, various regularization strategies, either implicit or explicit, have been proposed to mitigate overfitting (Mentch and Zhou, 2020; Agarwal et al., 2022). Recently, tree-based methods have been generalized to address the heterogeneous data from diverse data sources(Nasseri et al., 2022). For convenience, in the following we define a regression tree by two components: a set of leaves $\boldsymbol{Q} = \{Q_j\}_{j=1}^{J}$ and the associated parameter $\tau$. We can denote a causal tree by $T(X; \boldsymbol{Q}, \tau) = \sum_{j=1}^{J} \tau(Q_j) \mathrm{I}\{\boldsymbol{x} \in Q_j\}$, where $\mathrm{I}\{\cdot\}$ denotes the indicator function and $\tau(Q_j)$ denotes the casual effect of sub-area indexed by $Q_j$.

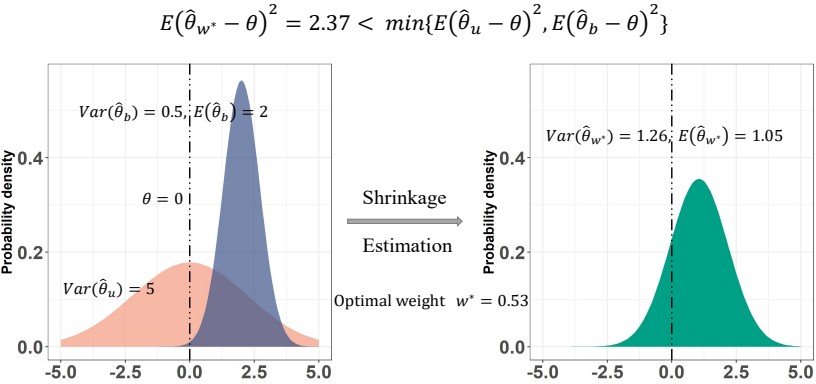

Figure 1: The probability density functions (pdfs) of the unbiased estimator (pink) and the biased estimator (blue) in the left panel and the pdf of the shrinkage (fused) estimator under the optimal weight $W^*$ (green) in the right panel. The vertical dashed line represents the true parameter value $\theta^* = 0$.

## 2.3 Shrinkage estimation

It is important to note that applying conventional methods, such as the generalized random forest (Athey et al., 2019), separately to trial data and observational data can readily lead to two estimators: the first is unbiased but may exhibit large variability, while the second is potentially biased but usually has a smaller variance due to the much larger amount of observational data. Therefore, the challenge becomes finding the optimal combination of an unbiased estimator and a biased estimator in the data fusion problem. To see this, suppose we have a parameter of interest $\theta \in \mathbb{R}$, an unbiased estimator $\hat{\theta}_u$, and a (potentially) biased estimator $\hat{\theta}_b$ of $\theta$, such that $\mathbb{E}(\hat{\theta}_u) = \theta$, $\mathbb{E}(\hat{\theta}_b) = \theta + b(\theta)$, $\mathrm{Var}(\hat{\theta}_u) = \sigma_u^2$, $\mathrm{Var}(\hat{\theta}_b) = \sigma_b^2$ and $\mathrm{Cov}(\hat{\theta}_u, \hat{\theta}_b) = 0$. Consider the family of estimators $\Lambda_w = \{\hat{\theta}_w | \hat{\theta}_w = w\hat{\theta}_b + (1-w)\hat{\theta}_u, 0 \leq w \leq 1\}$, then the mean square error (MSE) of its elements admits the following expansion:

$$\mathbb{E}(\hat{\theta}_w - \theta)^2 = (\sigma_b^2 + b^2(\theta) + \sigma_u^2)w^2 - 2\sigma_u^2 w + \sigma_u^2. \tag{1}$$

Minimizing (1) with respect to $w$, we can obtain the unique minimizer $w^* = \sigma_u^2/(\sigma_b^2 + b^2(\theta) + \sigma_u^2)$ and the gain of the optimal weighting over the single estimators $\hat{\theta}_u$ and $\hat{\theta}_b$ can be characterized by the following formula:

$$\mathbb{E}(\hat{\theta}_w^* - \theta)^2 = (1 - w^*)\sigma_u^2 = w^*(\sigma_b^2 + b^2(\theta)). \tag{2}$$

**Comment** The weighting strategy is akin to the classical James-Stein shrinkage estimation (Efron and Morris, 1973; Green and Strawderman, 1991) method, in which it is shown that a multivariate normal vector $\boldsymbol{Z}$ ($p \geq 3$), as a maximum likelihood estimator (MLE) of its population mean $\boldsymbol{\mu} = \mathbb{E}(\boldsymbol{Z})$, is not minimax optimal, and the MSE of the estimator $\boldsymbol{Z}$ can be reduced by shrinking it towards the zero vector $\boldsymbol{0}$ by some factor $0 < w < 1$. The zero vector can be viewed as a biased estimator of $\boldsymbol{\mu}$ with zero variance in their setting. In comparison, we replace the deterministic estimator with a (potentially) biased estimator $\hat{\theta}_b$: The larger the variance $\sigma_u^2$ of the unbiased estimator is compared to $b^2(\theta) + \sigma_b^2$, the more the fused estimator $\hat{\theta}_{w^*}$ will be shrunk towards the biased estimator that is less fluctuating. By doing so, one can efficiently mitigate the occurrence of significant estimation error in the unbiased estimator caused by its high variance, as unbiasedness alone *does not* guarantee reliable estimation performance with a limited sample size. Figure 1 illustrates a concrete example of the benefit provided by the shrinkage estimation, where $\theta = 0, \hat{\theta}_u \sim \mathrm{N}(\theta, 5)$ and $\hat{\theta}_b \sim N(\theta + 2, 0.5)$. The fused estimator $\hat{\theta}_{w^*}$ reduces over $50\%$ of the MSE compared with the unbiased estimator $\hat{\theta}_u$.

## 3 Methodology

In this section, we propose a new data fusion strategy, referred to as the Fused and Accurate Shrinkage Tree (FAST). We proceed in a bottom-up manner to provide a clear and intuitive illustration of the entire estimation: we will begin by applying the shrinkage estimation strategy for local data fusion within each sub-region of the feature space given by a pre-specified partition. Then, we propose a fused criterion that incorporates the information contained in the observational data via a simple re-scaling of the conventional criterion. Theoretical guarantees are established in Section 4.

### 3.1 Local fusion for the HTE estimation

Under a pre-specified partition $\boldsymbol{Q} = \{Q_j\}_{j=1}^J$ of the feature space, let $\mathcal{R}_j = \{i | i \in \mathcal{R}, \boldsymbol{X}_i \in Q_j\}$ and $\mathcal{O}_j = \{i | i \in \mathcal{O}, \boldsymbol{X}_i \in Q_j\}$ represent the sets of indices of the trial and observational sub-samples, respectively, that fall within the region $Q_j$. Let

$$\widetilde{Y} = \frac{YD}{e(\boldsymbol{X}, S)} - \frac{Y(1 - D)}{1 - e(\boldsymbol{X}, S)} \tag{3}$$

be transformed outcomes of all data, e.g., the transformed outcomes of $i$-th sample can be denoted by $\widetilde{Y}_i$. Then under Assumption 1, one can immediately show for the trial population with $S = 1$:

$$\begin{aligned} \mathbb{E}(YD|\boldsymbol{X}, S = 1) &= \mathbb{E}(Y(1)|\boldsymbol{X}, S = 1)\mathbb{E}(D|\boldsymbol{X}, S = 1) \quad \text{and} \\ \mathbb{E}(Y(1 - D)|\boldsymbol{X}, S = 1) &= \mathbb{E}(Y(0)|\boldsymbol{X}, S = 1)(1 - \mathbb{E}(D|\boldsymbol{X}, S = 1)), \quad \text{leading to} \end{aligned}$$

$$\mathbb{E}\left((\widetilde{Y}|\boldsymbol{X} = \boldsymbol{x}, S = 1\right) = \tau_1(\boldsymbol{x}) = \tau(\boldsymbol{x}). \tag{4}$$

Thus, $\hat{\tau}_u(Q_j) = (1/|\mathcal{R}_j|)\sum_{i \in \mathcal{R}_j} \widetilde{Y}_i$ is an unbiased estimator of $\mathbb{E}(Y(1) - Y(0)|\boldsymbol{X} \in Q_j, S = 1)$, which can be seen as a reasonable approximation of $\tau(Q_j)$ if $\boldsymbol{Q}$ segments the feature space properly such that $\tau(\boldsymbol{x})$ varies slowly in each sub-region $Q_j$. An estimator of $\mathrm{Var}(\hat{\tau}_u(Q_j))$ is given by $\hat{\sigma}_u^2(Q_j) = (1/|\mathcal{R}_j|(|\mathcal{R}_j| - 1)))\sum_{i \in \mathcal{R}_j}(\widetilde{Y}_i - \hat{\tau}_u(Q_j))^2$. In contrast, for the observational population, the conditional independence no longer holds and $\hat{\tau}_b(Q_j) = (1/|\mathcal{O}_j|)\sum_{i \in \mathcal{O}_j} \widetilde{Y}_i$ is a biased estimator concerning $\tau(Q_j)$, due to the presence of unmeasured confounding ($\boldsymbol{U}$) on the observational data.

It remains to estimate the weight $w^*(Q_j)$ composed of the tuple $(\sigma_u^2(Q_j), \sigma_b^2(Q_j), b^2(Q_j))$. The first term $\sigma_u^2(Q_j)$ can be estimated by $\hat{\sigma}_u^2(Q_j)$. To bypass the unmeasured confounding issue of the observational population, re-sampling techniques, such as the Bootstrap (Efron, 1979; Hall, 1992), can be applied to consistently estimate $\sigma_b^2(Q_j)$. However, in the causal data fusion setting, $\sigma_b^2(Q_j) = \mathrm{O}(|\mathcal{O}_j|^{-1})$ is expected to be of a smaller order term compared to $\sigma_u^2(Q_j) = \mathrm{O}(|\mathcal{R}_j|^{-1})$,

which is a consequence of the relative sample size between the trial and the observational data. Thus, one can just avoid estimating the negligible term $\sigma_b^2(Q_j)$. For the last term, $\widehat{b(Q_j)} = \hat{\tau}_b(Q_j) - \hat{\tau}_u(Q_j)$ serves as a natural estimator of the bias $b(Q_j)$. This leads to the following estimator of $w^*(Q_j)$ and the corresponding fused estimator

$$\hat{w}_{of}(Q_j) = \hat{\sigma}_u^2(Q_j)/(\hat{\sigma}_u^2(Q_j) + (\widehat{b(Q_j)})^2) \text{ and} \tag{5}$$

$$\hat{\tau}_{of}(Q_j) = \hat{w}_{of}(Q_j)\hat{\tau}_b(Q_j) + (1 - \hat{w}_{of}(Q_j))\hat{\tau}_u(Q_j). \tag{6}$$

A fused estimator of the HTE function $\tau(\cdot)$ under the partition $\boldsymbol{Q}$ can thus be defined as $\hat{\tau}_{\boldsymbol{Q}}(\boldsymbol{x}) = \sum_{j=1}^{J} \hat{\tau}_{of}(Q_j)\mathrm{I}\{\boldsymbol{x} \in Q_j\}$.

## 3.2 Adaptive fusion for segmentation

In order to obtain a tree-based partition $\boldsymbol{Q}$ designed for the fusion strategy (6), a split criterion is required, which is sufficient to be defined only at the root node given the recursive nature of the partitioning. We follow the honest estimation approach (Athey and Imbens, 2016) to prevent overfitting. Specifically, given a fraction $0 < r < 1$ (typically $r = 0.5$), $\lfloor rn \rfloor$ observations are sampled without replacement from the trial data of sample size $n$ for the tree structure estimation, while the rest of observations are used for local estimation of the HTE in each leaf node. Let the index sets of the trial data used for the partition and the HTE estimation be $\mathcal{R}^t$ and $\mathcal{R}^e$, respectively. We do not further split the observational data to reduce uncertainty, since we have already partitioned the trial data to avoid overfitting.

The conventional criterion for growing a regression tree chooses the index of the split variable and its split value at the root node by minimizing the following goodness-of-fit criterion

$$(\hat{q}, \hat{c}) = \arg\min_{\hat{q} \in [p], \hat{c} \in \mathbb{R}} \left( \sum_{i \in \widehat{\mathcal{R}}_L^t} \left( \widetilde{Y}_i - \hat{\tau}_u(\widehat{Q}_L, \mathcal{R}^t) \right)^2 + \sum_{i \in \widehat{\mathcal{R}}_R^t} \left( \widetilde{Y}_i - \hat{\tau}_u(\widehat{Q}_R, \mathcal{R}^t) \right)^2 \right), \tag{7}$$

where $\widehat{Q}_L = \{\boldsymbol{x}|\boldsymbol{x}_{\hat{q}} \leq \hat{c}\}$, $\widehat{\mathcal{R}}_L^t = \{i|i \in \mathcal{R}^t, \boldsymbol{X}_i \in \widehat{Q}_L\}$ and $\hat{\tau}_u(\widehat{Q}_L, \mathcal{R}^t) = (1/|\{i|i \in \mathcal{R}^t, \boldsymbol{X}_i \in \widehat{Q}_L\}|) \sum_{i \in \{i|i \in \mathcal{R}^t, \boldsymbol{X}_i \in \widehat{Q}_L\}} \widetilde{Y}_i$, and $\widehat{Q}_R$, $\widehat{\mathcal{R}}_R^t$ and $\hat{\tau}_u(\widehat{Q}_R, \mathcal{R}^t)$ can be defined correspondingly. Given a tree grown under (7), we fuse the trial data indexed by $\mathcal{R}^e$ and the observational data indexed by $\mathcal{O}$ at each leaf node according to (6) and refer to the resulting tree estimator as a **Shrinkage Tree** (ST). A direct modification of (7), which aligns more with the fused estimator at the leaf nodes, should be

$$(\hat{q}, \hat{c}) = \arg\min_{\hat{q} \in [p], \hat{c} \in \mathbb{R}} \left( \sum_{i \in \widehat{\mathcal{R}}_L^t} \left( \widetilde{Y}_i - \hat{\tau}_{of}(\widehat{Q}_L) \right)^2 + \sum_{i \in \widehat{\mathcal{R}}_R^t} \left( \widetilde{Y}_i - \hat{\tau}_{of}(\widehat{Q}_R) \right)^2 \right), \tag{8}$$

where $\hat{\tau}_{of}(\widehat{Q}_L) = \hat{w}_{of}(\widehat{Q}_L)\hat{\tau}_b(\widehat{Q}_L) + (1 - \hat{w}_{of}(\widehat{Q}_L))\hat{\tau}_u(\widehat{Q}_L, \mathcal{R}^t)$ and $\hat{\tau}_{of}(\widehat{Q}_R)$ is defined correspondingly. The replacement of the unbiased estimators in (7) with the fused estimators in (8) facilitates a goodness-of-fit criterion of the proposed fusion strategy.

Alternatively, (7) can be interpreted as minimizing the sum of the estimated MSEs of the unbiased estimators at the child nodes, if the two terms on the right-hand side of (7) are divided by the square of their respective sample sizes. By contrast, since the fused estimator $\hat{\tau}_{of}$ reduces variance by shrinking the original unbiased estimator to a potentially biased estimator, simply comparing the fused estimators with the outcomes of the trial data as in (8) fails to capture the variability at the child nodes. Instead, an appropriate criterion shall respect the MSE of the fused estimator. To this end, we introduce the following split criterion

$$(\hat{q}, \hat{c}) = \arg\min_{\hat{q} \in [p], \hat{c} \in \mathbb{R}} \left( (1 - \hat{w}_{of}(\widehat{Q}_L))\hat{\sigma}_u^2(\widehat{Q}_L, \mathcal{R}^t) + (1 - \hat{w}_{of}(\widehat{Q}_R))\hat{\sigma}_u^2(\widehat{Q}_R, \mathcal{R}^t) \right), \tag{9}$$

where $(1 - \hat{w}_{of}(\widehat{Q}_L))\hat{\sigma}_u^2(\widehat{Q}_L, \mathcal{R}^t)$ and $(1 - \hat{w}_{of}(\widehat{Q}_R))\hat{\sigma}_u^2(\widehat{Q}_R, \mathcal{R}^t)$ estimate the MSE of $\hat{\tau}_{of}(\widehat{Q}_L)$ and $\hat{\tau}_{of}(\widehat{Q}_R)$, respectively, according to formula (2). Compared to (7), the proposed criterion incorporates the additional information from the observational data into each node split in an adaptive manner by simply re-scaling the estimated MSE of the unbiased estimator.

**Comment** The criterion (9) offers the benefit of local adjustment, which can be intuitively justified. In sub-regions where the observational data exhibit moderate confounding biases, this criterion improves tree building by providing a sharper assessment of the variability of the fused estimator. On the other hand, for sub-regions where the observational data exhibit substantial confounding biases, the estimated weights of those sub-regions approach zero according to (5). In such cases, the criterion reduces to the conventional criterion (7), except for the standardization of the square of the sample size. It is worth mentioning that all the local adjustments achieved by applying this adaptive fusion strategy are data-driven, namely one can just avoid global modeling of the confounding bias function, which requires domain-specific knowledge of the observational studies. Additionally, it also enables the exclusion of the global impact of extremely large confounding biases of the observational data that only exist in certain sub-regions of the feature space.

We denote the partition obtained under criterion (9) as $\widehat{\boldsymbol{Q}}_{of} = \{\widehat{Q}_{of,1}, \widehat{Q}_{of,2}, \cdots, \widehat{Q}_{of,|\widehat{\boldsymbol{Q}}_{of}|}\}$, and the corresponding tree-based estimator of the HTE is defined as

$$\hat{\tau}_{fast}(\boldsymbol{x}) = \sum_{j=1}^{|\widehat{\boldsymbol{Q}}_{of}|} \hat{\tau}_{of}^{e}(\widehat{Q}_{of,j}) \mathrm{I}\{\boldsymbol{x} \in \widehat{Q}_{of,j}\}, \tag{10}$$

where the superscript "e" is to show that the RCT data used to construct the fused estimator at the leaf node is indexed by $\mathcal{R}^e$ and "fast" is an acronym for the name Fused and Accurate Shrinkage Tree, which is due to the data fusion nature of the criterion (9), the shrinkage-type leaf node estimator (6) and its accuracy in terms of the MSE.

## 3.3 Ensemble fusion

To reduce overfitting, improve robustness against outliers, and enhance generalization, we introduce the bagged version (Hastie et al., 2009) of the FAST, referred to as the rfFAST, as follows: We randomly draw index sets $\mathcal{R}^*$ of size $n$ and $\mathcal{O}^*$ of size $m$, separately from $\mathcal{R}$ and $\mathcal{O}$ with replacement. We repeat the process $B$ times, resulting in $\{\mathcal{R}^{*,(b)}, \mathcal{O}^{*,(b)}\}_{b=1}^{B}$. Then, $B$ estimators $\hat{\tau}_{fast}^{*,(b)}(\boldsymbol{x})$ can be calculated based on the trial data indexed by $\mathcal{R}^{*,(b)}$ and the observational data index by $\mathcal{O}^{*,(b)}$. We then define $\hat{\tau}_{rffast}(\boldsymbol{x}) = (1/B) \sum_{b=1}^{B} \hat{\tau}_{fast}^{*,(b)}(\boldsymbol{x})$. For the construction of the prediction intervals, see Zhang et al. (2020).

## 4 Theoretical guarantee

In this section, we formally establish the benefits of the proposed split criterion (9) compared with the conventional criterion (7). To present the theoretical result, we first pose the following regularity conditions that are standard in literature (see e.g., Györfi et al., 2002 and Scornet et al., 2015).

**Assumption 2.** *(i) There exists a positive constant $\lambda < \infty$ such that $\mathbb{E}\{\exp(\lambda \tilde{Y}^2)|S = i\} < \infty$ for $i = 0, 1$. (ii) There exists positive constants $\sigma_{\min} < \infty$ such that $\sigma_{\min}^2 < \mathrm{Var}(\tilde{Y}|\boldsymbol{X} = \boldsymbol{x}, S = 0)$ for any $\boldsymbol{x} \in \mathcal{X}$.*

**Theorem 1** (MSE reduction of the proposed split criterion). *Let $\theta = (q, c)$ and $\Theta = [p] \times \mathbb{R}$. Suppose the node that needs to be partitioned is $Q_j$, under which the sample sizes of the trial data and observational data are $n_j$ and $m_j$, respectively. Let $M(\theta)$ and $M_{of}(\theta)$ be the sum of MSEs of the conventional HTE estimator and the fused HTE estimator on the two child nodes of $Q_j$ split by $\theta$, respectively. Denote $b_{\max} = \sup_{\boldsymbol{x} \in Q_j} |\{\mathbb{E}(\tilde{Y}|\boldsymbol{X} = \boldsymbol{x}, S = 0) - \mathbb{E}(\tilde{Y}|\boldsymbol{X} = \boldsymbol{x}, S = 1)\}|$. Let $\hat{\theta}$ be the solution of the conventional split criterion (7) and $\hat{\theta}_{of}$ be the solution of the proposed split criterion (9). Under Assumptions 1-2, we have*

*(i) For any $\theta \in \Theta$,*

$$\frac{M_{of}(\theta)}{M(\theta)} - 1 \leq -\frac{\sigma_{\min}^2}{\sigma_{\min}^2 + n_j b_{\max}^2}. \tag{11}$$

*(ii) With probability at least $1 - C_1 e^{-t}$ for some positive constant $C_1 < \infty$, it holds that*

$$M(\hat{\theta}) - M(\theta^*) \le C_2 \frac{t + \log(pn_j)\log^4(n_j)}{n_j}, \tag{12}$$

$$and \; M_{of}(\hat{\theta}_{of}) - M_{of}(\theta^*_{of}) \le C_3 \left( \frac{t + \log(pn_j)\log^4(n_j)}{m_j} + \frac{t + \log(pn_j)\log^4(n_j)}{n_j} \right), \tag{13}$$

*for some positive constant $C_2, C_3 < \infty$, where $\theta^*$ and $\theta^*_{of}$ are oracle splits defined as*

$$\theta^* = \arg\min_{\theta \in \Theta} M(\theta) \;\; and \;\; \theta^*_{of} = \arg\min_{\theta \in \Theta} M_{of}(\theta).$$

In the above theorem, the (i) part establishes a uniform MSE reduction result for any split choice $\theta \in \Theta$ of the proposed split criterion (9). As revealed in (11), the criterion (9) leads to larger MSE reduction on the nodes with a larger variance of $\tilde{Y}$ and less bias of the observational data. In addition, the upper bound in (11) decreases as the node sample size $n_j$ decreases, implying that our proposed criterion leads to increasing relative benefits as the tree grows deeper. Besides, in the (ii) part we present non-asymptotic bounds for the discrepancies between the MSEs under the empirically estimated splits and the oracle splits, showing that the MSEs under the estimated splits can achieve a fast convergence rate. As a direct consequence of Theorem 1, the consistency of our final HTE estimator (10) can be established, since it is known from Scornet et al. (2015) and Athey et al. (2019) that the conventional tree-based estimator using only the trial data is mean-squared consistent, and our proposed method leads to a reduced MSE.

**Proposition 1** (Consistency of $\hat{\tau}_{fast}$). *For almost every $\boldsymbol{x} \in [-1, 1]^p$, we have $\hat{\tau}_{fast}(\boldsymbol{x}) \to \tau(\boldsymbol{x})$ in probability as $n, m \to \infty$.*

## 5 Experiments

In this section, we demonstrate the results of a series of experiments to answer the following two questions: (i) Whether the proposed method can effectively alleviate the impact of confounding bias of observational data and limited sample size of trial data; (ii) Whether the techniques we proposed including local fusion in tree leaves and adaptive fusion in partitioning are valid, respectively.

In consequence, we conducted experiments on both simulated and real-world datasets to verify the effectiveness of our method. We evaluated our method against both traditional tree-based and data fusion-based casual methods. The former includes the classical Transformed Outcome Honest Tree (HT) Athey and Imbens (2016) and its ensemble version Generalized Random Forest (GRF) Athey et al. (2019). The latter includes the simplest fusion estimator (SF) training both trial data and observational data together without distinction and the KPS estimators Kallus et al. (2018). In order to facilitate better comparison and understanding of our proposed method, we demonstrate three versions: the simple implementation, Shrinkage Tree (ST), described in Section 3.1; the improved version, Fused and Accurate Shrinkage Tree (FAST), described in Section 3.2; and its final ensemble version rfFAST described in Section 3.3. The results of each simulation experiment were based on $B = 100$ replications. The ensemble size for all the ensemble estimators was set to 100. For the tree estimators, the minimum number of observations required to be at a leaf node was set to 5 and the maximum depth of the tree was set to 10.

### 5.1 Simulation

We conducted two sets of simulation experiments to evaluate the finite sample performance of the fused estimator and various baseline estimators. In both experiments, we first generated the pre-treatment covariates $\boldsymbol{X} = (X_1, X_2, \cdots, X_p)^T$ from $\mathrm{Uniform}[-1, 1]^p$ and the unobserved variable $U$ from $\mathrm{N}(0, 1)$. Then, we generated the potential outcomes by $Y(d) = d\tau(\boldsymbol{X}) + \sum_{j=1}^{p} X_j + 1.5U + \epsilon(d)$, where $\tau(\boldsymbol{X}) = 1 + X_1 + X_1^2 + X_2 + X_2^2$ and $\epsilon(d) \sim \mathrm{N}(0, 1)$ for $d = 0, 1$. Thus The treatment assignments for the trial sample of size $n$ and the observational sample of size $m$ were generated as follows: $D|(\boldsymbol{X}, U, S = 1) \sim \mathrm{Ber}(0.5)$ and $D|(\boldsymbol{X}, U, S = 0) \sim \mathrm{Ber}(1/(1 + \exp(-\beta U - 0.5X_1)))$. Thus, the magnitude of $\beta$ controls the strength of the unmeasured confounding: a larger $\beta$ leads to a larger confounding bias. The test data $X_{test,j}$ for $1 \le j \le p$ were generated from $\mathrm{Uniform}(-1, 1)$ with sample size 1000.

In the first experiment, we aim to verify the effectiveness of the proposed data fusion strategy via an ablation study. We compared the robustness of the ST and the FAST against different levels of confounding bias parameter $\beta$. Two baselines were considered: (i) the HT using only the trial data and (ii) the SF estimator obtained by directly merging all the available data and constructing a Fit-Based Causal Tree (Athey and Imbens, 2016). We set the sample sizes of the trial data and the observational data be $n = 200$ and $m = 2000$, respectively, the dimension of covariates $p = 5$ and $\beta \in \{0.1c | c \in \mathbb{N}, c \leq 19\}$. The following three conclusions could be drawn from Figure 2: (1) When confounding bias in observational data was small, the simple fusion (SF) strategy can effectively improve the model performance. But when it became large, the SF was very vulnerable to confounding bias in observational data; (2) Even with the increase of $\beta$, both ST and FAST consistently showed resistance to confounding bias; (3) FAST was significantly better than other methods including ST, which verified the effectiveness of our proposed split criterion (9) numerically.

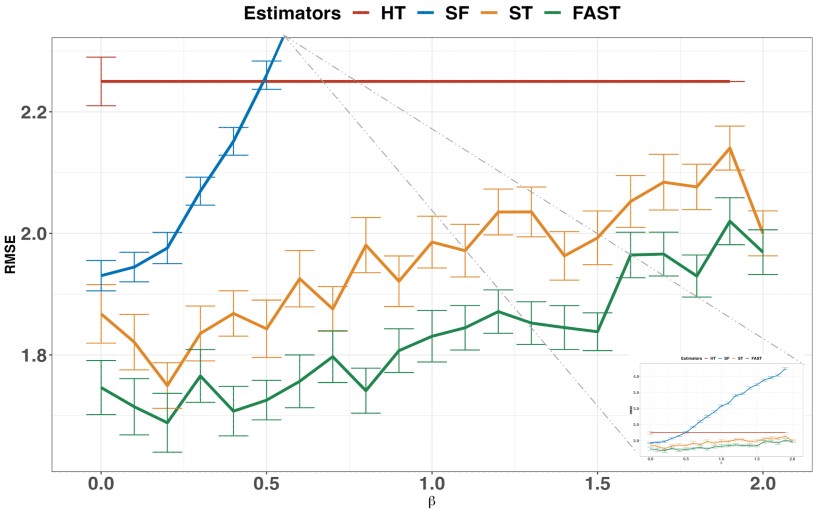

Figure 2: The averaged root mean square error (RMSE) (mean with s.e. error bars) of the estimators on simulation datasets with different levels of the confounding bias parameter $\beta$.

In the second experiment, we evaluated the RMSEs with respect to different $n$ and $\beta$. We set $m = 2000$ and $p = 5$. We included seven estimators in the analysis: The first two estimators were calculated purely based on the trial data: (i) the Transformed Outcome Honest Tree (HT) (Athey and Imbens, 2016) and (ii) the Generalized Random Forest (GRF) (Athey et al., 2019). The rest estimators were calculated using different data fusion strategies: (iii) the Shrinkage Tree (ST) estimator,(iv) the Fused and Accurate Shrinkage Tree (FAST) estimator, (v& vi) the KPS estimators (Kallus et al., 2018) with a parametric (OLS) estimator and a non-parametric (Random Forest) specification of the confounding function, respectively and (vii) the bagged FAST estimator (rfFAST).

Table 1 reports the RMSEs of the seven estimators, conveying a good estimation accuracy of both the FAST and its ensemble version rfFAST. Among the three individual estimators, the ST and FAST, exhibited superior performance compared to the HT, and the FAST outperformed the ST. These relative performances provided support for the FAST approach compared to the classical honest regression tree, the proposed split criterion (9), and the shrinkage estimation strategy (6), which are implemented progressively. Among the three ensemble estimators, the rfFAST estimator demonstrated the best performance among all the six combinations of the trial sample size $n$ and the confounding bias parameter $\beta$. On the other hand, the performance of the KPS estimators appeared to be less stable. The $\text{KPS}_{ols}$ outperformed the GRF only when the trial sample size was relatively large ($n = 200$). Under the non-parametric specification of the confounding function, the $\text{KPS}_{RF}$ did not gain benefit from incorporating the observational data and was consistently inferior to the baseline estimator GRF.

Table 1: The averaged RMSE (standard error in parentheses) of the estimators with respect to the trial sample size $n$ and the confounding bias parameter $\beta$. The best performance is marked in **bold**.

| $n$ | $\beta$ | HT | ST | FAST | GRF | $\text{KPS}_{ols}$ | $\text{KPS}_{RF}$ | rfFAST |
|---|---|---|---|---|---|---|---|---|
| | 0.5 | | 1.89 (0.06) | 1.84 (0.06) | | 1.33 (0.04) | 1.73 (0.03) | **0.84** (0.02) |
| 100 | 1.0 | 2.28 (0.06) | 1.90 (0.05) | 1.85 (0.05) | 1.19 (0.02) | 1.29 (0.04) | 1.65 (0.03) | **0.89** (0.02) |
| | 2.0 | | 2.05 (0.05) | 2.02 (0.04) | | 1.28 (0.04) | 1.71 (0.03) | **0.98** (0.02) |
| | 0.5 | | 1.87 (0.04) | 1.71 (0.04) | | 0.96 (0.02) | 1.56 (0.02) | **0.73** (0.01) |
| 200 | 1.0 | 2.20 (0.04) | 1.98 (0.04) | 1.83 (0.04) | 1.12 (0.01) | 0.97 (0.03) | 1.59 (0.02) | **0.84** (0.02) |
| | 2.0 | | 2.08 (0.03) | 1.97 (0.03) | | 1.01 (0.02) | 1.57 (0.03) | **0.92** (0.02) |

## 5.2 Real-world data

In this sub-section, we report an analysis of the Tennessee Student/Teacher Achievement Ratio (STAR) Experiment (Krueger, 1999) to demonstrate the proposed FAST for the HTE estimation. We aim at quantifying the treatment effect of the class size on the student's academic achievement.

**Data description** The STAR Experiment was a randomized controlled trial conducted in the late 1980s. Students were randomly assigned to one of the two types of classes during the first school year: $D = 1$ for small classes containing $13 - 17$ pupils and $D = 0$ for regular classes containing $22 - 25$ pupils. The outcome $Y$ is the average of the listening, reading, and math standardized tests at the end of first grade. The vector of covariates $X$ includes gender, race, birth month, birthday, birth year, free lunch given or not, and teacher id. This made a universal sample of 4218 students, among which 2413 were randomly assigned to regular-size classes ($D = 0$) and 1805 to small classes ($D = 1$).

**Ground-truth** In practice, the ground-truth $\tau(\cdot)$ is not accessible, so we replaced it with an estimate calculated by a generalized random forest (Athey et al., 2019) based on all the 4218 observations.

**Construction of the trial, observational and test data** Following Kallus et al. (2018), we introduced confounding bias by splitting the population over a variable which is known to strongly affect the observed outcome $Y$ (Krueger, 1999): rural or inner-city ($U = 1$, 2811 students) and urban or suburban ($U = 0$, 1407 students). The trial data were generated by randomly sampling a fraction $h$ of the students with $U = 1$, where $h$ ranges from 0.1 to 0.5. The observational data were constructed as follows: From students with $U = 1$, we took the controls ($D = 0$) that were not sampled in trial data, and the treated ($D = 1$) whose outcomes were in the lower half of outcomes among students with $D = 1$ and $U = 1$; From students with $U = 0$, we took all of the controls ($D = 0$), and the treated ($D = 1$) whose outcomes were in the lower half of outcomes among students with $D = 1$ and $U = 0$. The test data consisted of a held-out sub-sample of all the observations in the universal sample excluding the trial data.

**Results** We compared the performance of the rfFAST with various baseline estimators. In particular, the NF and the SF estimators were constructed using the Random Forest regressor. The NF estimator utilized only trial data, while the SF estimator utilized both trial data and observational data together without distinction. As shown in Figure 3, the proposed rfFAST method consistently outperformed other estimators.

## 6 Discussion

This paper explores the estimation of heterogeneous treatment effects (HTE) within the framework of causal data fusion. Drawing inspiration from the classical James-Stein shrinkage estimation (Green and Strawderman, 1991) approach, the authors introduce a new method called Fused and Accurate Shrinkage Tree (FAST) that effectively incorporates observational data in both feature

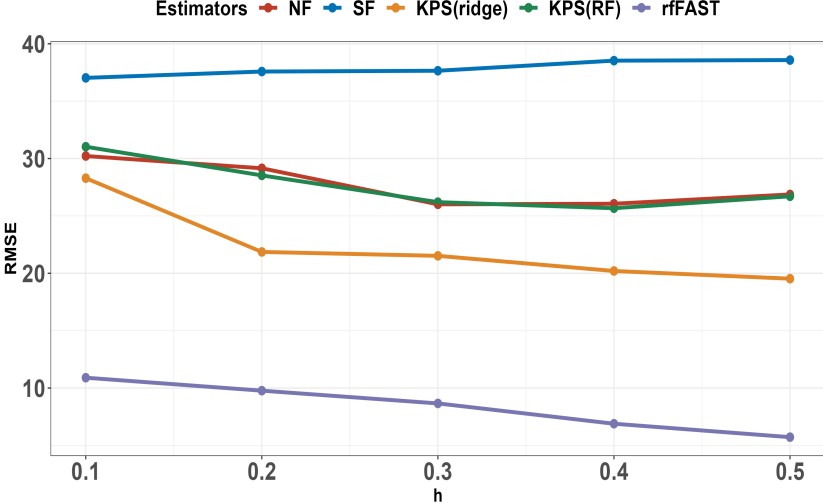

Figure 3: The RMSEs of the five estimators with respect to different sample sizes of the trial data, reflected by the fraction parameter $h$. A large $h$ means a large trial sample size.

space segmentation and leaf node value estimation. This new approach is shown to outperform existing data fusion methods via numerical experiments.

The above estimation framework can be generalized to any data fusion problem if there exists an unbiased estimator and a biased estimator of some functions of interest. It would be worthwhile to explore the combination of the FAST method with other ensemble methods, such as the boosting and the grf-style (Athey et al., 2019) bagging, in addition to Breiman-style (Breiman, 2001) bagging used in rfFAST. Moreover, extending the framework to handle time-series observational data would be an interesting direction for future research. Additionally, investigating statistical inference under the proposed fusion framework would also be valuable.

## Acknowledgements

This work is supported by Ant Group through Ant Research Intern Program, and funded by National Natural Science Foundation of China Grants 92046021, 12071013403 and 12026607. We thank the editors and five anonymous referees for their constructive suggestions and comments.

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

# APPENDIX

## A    Additional figures

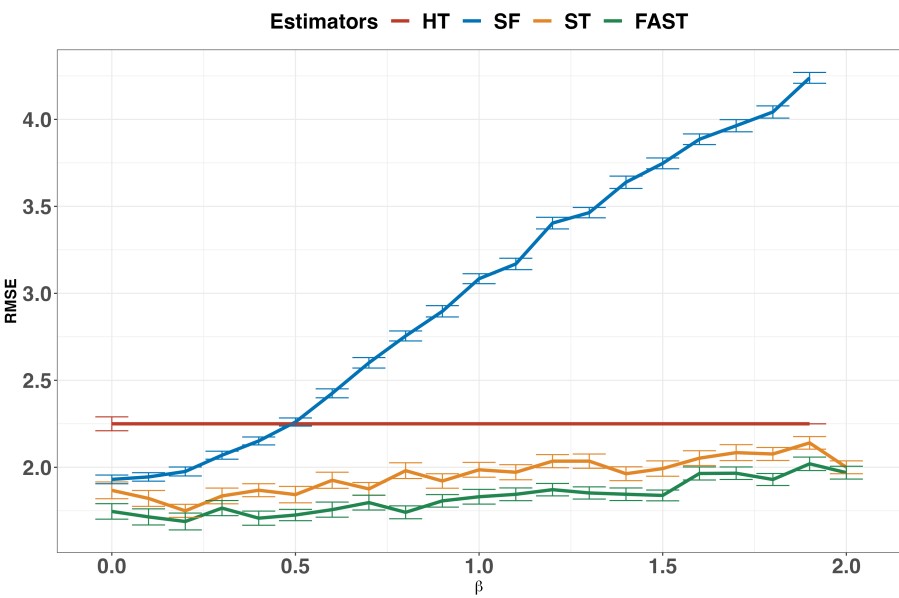

Figure 4: The averaged root mean square error (RMSE) (mean with $2\times$s.d. error bars) of each algorithm on multiple simulation datasets with different levels of the confounding bias parameter $\beta$.

## B    Pre-processing of the real-world data

In the STAR dataset, each of the pre-treatment covariate $X_j$ ($1 \leq j \leq p$) was standardized to a range of $-1$ to $1$, and the outcome variable $Y$ was standardized to a range of $0$ to $100$.

## C    Proof of Theorem 1

The proof follows the similar arguments as in Györfi et al. (2002) and Scornet et al. (2015). It is sufficient to show the result at the root node given the recursive nature of the partitioning. We will use the following notations in the sequel. We denote $\mathbb{E}_T, \mathbb{P}_T$ and $\mathbb{E}_O, \mathbb{P}_O$ as the expectation and probability under trial data and observational data, respectively. We let $Z = (X, \tilde{Y})$. For any $q \in [p]$ and $c \in \mathbb{R}$, let $\theta = (q, c)$ and the corresponding two partitioned notes are denoted as $Q_L(\theta) = \{x|x_q \leq c\}$ and $Q_R(\theta) = \{x|x_q > c\}$. The parameter space of $\theta$ is denoted as $\Theta = [p] \times \mathbb{R}$. Let $\mu_L$ and $\mu_R$ be the predictions for $Y$ on $Q_L(\theta)$ and $Q_R(\theta)$, respectively and denote $\tau = (\tau_L, \tau_R)$. Let $M^i(\theta)$ and $M^i_{of}(\theta)$ be the MSEs of the conventional HTE estimator and the fused HTE estimator on the child nodes of $Q_i(\theta)$, respectively, for $i \in \{R, L\}$.

**(i).** For any $\theta \in \Theta$, according to Equation (2) in the main paper we have

$$M^i_{of}(\theta) = (1 - w_i(\theta))M^i(\theta),$$

for $i \in \{R, L\}$, where the weight $w_i(\theta)$ satisfies

$$w_i(\theta) \asymp \frac{\sigma_u^2(Q_i)/n}{\sigma_u^2(Q_i)/n + b^2(\theta)},$$

by Equation (5) in the main paper, which is lower bounded by $\frac{\sigma_{\min}^2}{\sigma_{\min}^2 + nb_{\max}^2}$, where $\sigma_{\min}^2 <$ $\mathrm{Var}(\tilde{Y}|\boldsymbol{X} = \boldsymbol{x}, S = 0)$ and $b_{\max} = \sup_{\boldsymbol{x} \in Q_j} |\{\mathbb{E}(\tilde{Y}|\boldsymbol{X} = \boldsymbol{x}, S = 0) - \mathbb{E}(\tilde{Y}|\boldsymbol{X} = \boldsymbol{x}, S = 1)\}|$

Therefore, we conclude that

$$\frac{M_{of}(\theta)}{M(\theta)} - 1 \leq -\frac{\sigma_{\min}^2}{\sigma_{\min}^2 + nb_{\max}^2},$$

which reveals the MSE reduction effect of the proposed split criterion.

**(ii).** The proof includes two parts. In Part 1, we will derive the bounds for the discrepancies between the MSEs under the empirically estimated split and the oracle split under the conventional criterion, and in Part 2 the similar results under the proposed split criterion.

**Part 1.** We define the following criterion function:

$$\ell_n(\theta, \tau, \mathcal{R}_n^t) = \frac{1}{n} \sum_{i=1}^n \left\{ (\tilde{Y}_{0,i} - \tau_L)^2 I\{X_{0,i} \in Q_L(\theta)\} + (\tilde{Y}_{0,i} - \tau_R)^2 I\{X_{0,i} \in Q_R(\theta)\} \right\}$$

$$=: \ell_n^L(\theta, \tau_L, \mathcal{R}_n^t) + \ell_n^R(\theta, \tau_R, \mathcal{R}_n^t).$$

For $i \in \{L, R\}$, let

$$\mathcal{L}^i(\theta, \tau_i) = \mathbb{E}_T \left\{ \ell_n^i(\theta, \tau_i, \mathcal{R}_n^t) \right\} \quad \text{and} \quad \mathcal{L}(\theta, \tau) = \mathcal{L}_n^L(\theta, \tau_L) + \mathcal{L}_n^R(\theta, \tau_R) \tag{14}$$

Then $\mathcal{L}^i(\theta, \tau_i)$ represents the MSE of $\tau_i$ on the region $Q_i(\theta)$. For a given split $\theta = (q, c)$, it is straightforward to see that the optimal $\tau(\theta) = (\tau_L(\theta), \tau_R(\theta))$ is given by

$$\tau_i(\theta) = \arg\min_{\tau_i \in \mathbb{R}} \ell_n^i(\theta, \tau_i, \mathcal{R}_n^t) = \mathbb{E}_n \left\{ \tilde{Y}_0 | X_0 \in Q_i(\theta) \right\}$$

for $i \in \{L, R\}$, which is the sample mean of $Y$ on the region $Q_i(\theta)$. Therefore, by the definition of $M^i(\theta)$, it holds that $\mathcal{L}^i(\theta, \tau_i(\theta)) = M^i(\theta)$ for $i \in \{L, R\}$. The optimal split $\theta_0 = (q_0, c_0)$ on the population level is defined via minimizing the profiled criterion function:

$$(q_0, c_0) = \arg\min_{q \in [p], c \in \mathbb{R}} \left\{ M^L(\theta) + M^R(\theta) \right\} = \arg\min_{q \in [p], c \in \mathbb{R}} M(\theta).$$

Define $M_n^i(\theta) = \ell_n^i(\theta, \tau_i(\theta), \mathcal{R}_n^t)$ for $i \in \{L, R\}$ and the empirical optimal split $\widehat{\theta} = (\widehat{q}, \widehat{c})$ is defined via minimizing the sample criterion function:

$$(\widehat{q}, \widehat{c}) = \arg\min_{q \in [p], c \in \mathbb{R}} \left\{ M_n^L(\theta) + M_n^R(\theta) \right\} =: \arg\min_{q \in [p], c \in \mathbb{R}} M_n(\theta).$$

**Step 1** (Main error decomposition).

Now we will bound $M(\widehat{\theta}) - M(\theta_0)$, which represents the discrepancy of the MSEs of the oracle and empirical split. To apply empirical process theories for stochastic error analysis, we will use a truncation argument. We let $M_{n,\beta_n}^i(\theta, \pi_i, \mathcal{R}_n^t) = \mathbb{E}_n(T_{\beta_n}\tilde{Y} - T_{\beta_n}\pi_i(\theta))^2 I(X \in Q_i(\theta))$ and $M_{\beta_n}^i(\theta) = \mathbb{E}_T \left\{ M_{n,\beta_n}^i(\theta, \pi_i(\theta), \mathcal{R}_n^t) \right\}$, where $T_{\beta_n}x =: (|x| \wedge \beta_n)\text{sign}(x)$ for any $\beta_n > 0$. Correspondingly, let $M_{\beta_n}(\theta) = M_{\beta_n}^L(\theta) + M_{\beta_n}^R(\theta)$ and $M_{n,\beta_n}(\theta) = M_{n,\beta_n}^L(\theta) + M_{n,\beta_n}^R(\theta)$. Then we have the following error decomposition:

$$\begin{aligned}
0 &< M(\widehat{\theta}) - M(\theta_0) \\
&= M(\widehat{\theta}) - M_{\beta_n}(\widehat{\theta}) - M(\theta_0) + M_{\beta_n}(\theta_0) \\
&\quad + M_{\beta_n}(\widehat{\theta}) - M_{\beta_n}(\theta_0) - 2M_{n,\beta_n}(\widehat{\theta}) + 2M_{n,\beta_n}(\theta_0) \\
&\quad + 2M_{n,\beta_n}(\widehat{\theta}) - 2M_n(\widehat{\theta}) - 2M_{n,\beta_n}(\theta_0) + 2M_n(\theta_0) \\
&\quad + 2M_n(\widehat{\theta}) - 2M_n(\theta_0) \\
&=: S_{1,n} + S_{2,n} + S_{3,n} + S_{4,n}.
\end{aligned}$$

By the definition of $\widehat{\theta}$, we have $S_{4,n} \leq 0$. In following steps, we will bound $S_{1,n}, S_{2,n}$ and $S_{3,n}$, respectively. The truncation level $\beta_n$ is chosen as $\beta_n = \beta_0 \log(n)$ for $\beta_0 \geq 2\sigma_Y$.

**Step 2** (Bounding $S_{1,n}$).

For any $\theta$, it holds that

$$
\begin{aligned}
M^i(\theta) - M^i_{\beta_n}(\theta) =& \mathbb{E}_T \left\{ (\tilde{Y} - \widehat{\tau}_i(\theta))^2 - (T_{\beta_n}\tilde{Y} - T_{\beta_n}\widehat{\tau}_i(\theta))^2 I\{X \in Q_i(\theta)\} \right\} \\
=& \mathbb{E}_T \left\{ (\tilde{Y} - T_{\beta_n}\tilde{Y})(\tilde{Y} + T_{\beta_n}\tilde{Y} - 2\widehat{\pi}_i(\theta)) I\{X \in Q_i(\theta)\} \right\} \\
& + \mathbb{E}_T \left\{ (T_{\beta_n}\widehat{\pi}_i(\theta) - \widehat{\pi}_i(\theta))(T_{\beta_n}\tilde{Y} + T_{\beta_n}\widehat{\pi}_i(\theta) - 2\widehat{\pi}_i(\theta)) I\{X \in Q_i(\theta)\} \right\} \\
=:& S_{5,n} + S_{6,n}.
\end{aligned}
$$

For $T_{1,n}$, by Cauchy-Schwarz inequality we have

$$
|S_{5,n}| \leq \sqrt{\mathbb{E}_T(\tilde{Y} - T_{\beta_n}\tilde{Y})^2} \sqrt{\mathbb{E}_T(\tilde{Y} + T_{\beta_n}\tilde{Y} - 2\widehat{\pi}_i(\theta))^2} \lesssim \sqrt{\mathbb{E}_T(\tilde{Y} - T_{\beta_n}\tilde{Y})^2},
$$

where the second inequality is because $\mathbb{E}_T(\tilde{Y}^2) \leq \infty$ and $\mathbb{E}_T\left\{\widehat{\pi}_i^2(\theta)\right\} \leq \mathbb{E}_T(\tilde{Y}^2)/|Q_i(\theta)|$. Since

$$
I(|\tilde{Y}| > \beta_n) \leq \frac{\exp(\sigma_Y|Y|^2/2)}{\sigma_Y \beta_n^2/2},
$$

therefore,

$$
|T_{1,n}| \lesssim \sqrt{\mathbb{E}_T(\tilde{Y} - T_{\beta_n}\tilde{Y})^2} \leq \sqrt{\mathbb{E}_T\left\{|Y|^2 \frac{\exp(\sigma_Y|Y|^2/2)}{\sigma_Y \beta_n^2/2}\right\}} \leq \sqrt{\frac{2}{\sigma_Y}\mathbb{E}_T \exp(\sigma_Y|Y|^2)} \exp(-\frac{\sigma_Y \beta_n^2}{4}).
$$

Since $\mathbb{E}_T \exp(\sigma_Y|Y|^2) < \infty$ and $\beta_n = \beta_0 \log(n)$, we conclude that $|S_{5,n}| \lesssim \frac{1}{n}$. With the same argument, we have $S_{6,n} \lesssim \frac{1}{n}$, implying that

$$
M(\theta) - M_{\beta_n}(\theta) \lesssim \frac{1}{n} \tag{15}
$$

for any $\theta \in \Theta$. Therefore, the truncation error $S_{1,n} \lesssim \frac{1}{n}$.

**Step 3** (Bounding $S_{2,n}$).

Let $M_{N,of} = \left\{ f = (T_{\beta_n}\tilde{Y} - T_{\beta_n}\pi) I(X \in Q_i(\theta)) : \theta = (q,c) \in [p] \times \mathbb{R} \right\}$. By applying Lemma 2 we obtain

$$
\mathcal{N}_1(\delta, M_{N,of}, z_1^n) \leq (pn)^2 \left(\frac{c\beta_n}{\delta}\right)^4,
$$

where $z_1^n$ is any set $\{z_1, \cdots, z_n\} \in [0,1]^p \times \mathcal{Y}$ and $c > 0$ is a universal constant. It follows from Lemma 1 that

$$
\mathbb{P}_T \left\{ \exists \theta \in \Theta : |M_{\beta_n}(\theta) - M_{n,\beta_n}(\theta)| \geq \frac{1}{2}(\alpha + \gamma + M_{\beta_n}(\theta)) \right\}
$$

$$
\leq 28(pn)^2 \left(\frac{80c\beta_n^2}{\gamma}\right)^4 \exp\left(-\frac{\alpha n}{1284\beta_n^4}\right)
$$

$$
\lesssim \exp\left(-\frac{\alpha n}{\beta_n^4} + \log(pn) - \log(\gamma)\right).
$$

Taking $\gamma = 1/n$ and $\alpha = (t + \log(pn))\beta_n^4/n$ implies that with probability at least $1 - C_1 e^{-t}$ for some universal constant $C_1 > 0$,

$$
\forall \theta \in \Theta, |M_{\beta_n}(\theta) - 2M_{n,\beta_n}(\theta)| \lesssim \frac{t + \log(pn)\log^4(n)}{n}. \tag{16}
$$

Therefore, we conclude that with probability at least $1 - C_1 e^{-t}$, the stochastic error $S_{2,n} \lesssim \left\{ t + \log(pn)\log^4(n) \right\}/n$.

**Step 4** (Bounding $S_{3,n}$). According to (15), we have

$$\forall \theta \in \Theta : \mathbb{E}_T \left\{ M_{n,\beta_n}(\theta) - M_n(\theta) \right\} \lesssim \frac{1}{n}$$

Since $\tilde{Y}$ is sub-Gaussian by assumption, it is straightforward to see that $(T_{\beta_n}\tilde{Y} - T_{\beta_n}\pi_i(\theta))^2 I(X \in Q_i(\theta))$ and $(\tilde{Y} - \pi_i(\theta))^2 I(X \in Q_i(\theta))$ are sub-exponential for $i \in \{L, R\}$ . Suppose $\left\| (T_{\beta_n}\tilde{Y} - T_{\beta_n}\pi_i(\theta))^2 I(X \in Q_i(\theta)) \right\|_{\psi_1} \leq \sigma_0$ and $\left\| (\tilde{Y} - \pi_i(\theta))^2 I(X \in Q_i(\theta)) \right\|_{\psi_1} \leq \sigma_0$ for all $\theta \in \Theta$, where $\|\cdot\|_{\psi_1}$ is the sub-exponential norm operator. By applying Bernstein's inequality, for any $s > 0$, we have

$$\mathbb{P}_T \left\{ \left| M_{n,\beta_n}^i(\theta) - M_n^i(\theta) - \mathbb{E}_T \left\{ M_{n,\beta_n}^i(\theta) - M_n^i(\theta) \right\} \geq s \right| \right\}$$
$$\leq 2 \exp \left( -c \min \left( \frac{ns^2}{\sigma_0^2}, \frac{ns}{\sigma_0} \right) \right),$$

for $i \in \{R, L\}$, where $c > 0$ is a universal constant. Taking $s = \frac{\sigma_0 t}{cn} = C_2 t$, for any $t \geq 0$ we obtain

$$\mathbb{P}_T \left\{ \left| M_{n,\beta_n}^i(\theta) - M_n^i(\theta) - \mathbb{E}_T \left\{ M_{n,\beta_n}^i(\theta) - M_n^i(\theta) \right\} \geq C_2 t \right| \right\} \leq 2 \exp(-t) \qquad (17)$$

for any $n > t/c$. Since the above result holds for any $\theta \in \Theta$, we conclude that for any $t > 0$, with probability at least $1 - 4e^{-t}$, we have $S_{3,n} \lesssim (t+1)/n$.

Combining the results on $S_{1,n}, S_{2,n}$ and $S_{3,n}$, we conclude that for any $t > 0$, with probability at least $1 - C_3 e^{-t}$, it holds that

$$\mathcal{L}_i(\widehat{\theta}, \widehat{\pi}_i(\widehat{\theta})) - \mathcal{L}_i(\theta_0, \pi(\theta_0)) \lesssim \frac{t + \log(pn) \log^4(n)}{n}, \qquad (18)$$

for some universal constants $C_3, C_4 > 0$.

**Part 2.** The proposed scale criterion can reformulated as follows. For $i \in \{L, R\}$, let

$$F_{0,i}(\theta) = \{1 - w_i(\theta)\} (\tilde{Y}_0 - \tau_{0,i}(\theta))^2 I(X_0 \in Q_i(\theta))$$
$$F_{1,i}(\theta) = w_i(\theta)(\tilde{Y}_1 - \tau_{1,i}(\theta))^2 I(X_1 \in Q_i(\theta)) \text{ and}$$

where $\tau_{0,i}(\theta) = \mathbb{E}_n(\tilde{Y}_0 | X_0 \in Q_i(\theta))$ and $\tau_{1,i}(\theta) = \mathbb{E}_m(\tilde{Y}_1 | X_1 \in Q_i(\theta))$, and

$$w_i(\theta) = \sigma_u^2(Q_i(\theta)) / \left\{ \sigma_u^2(Q_i(\theta)) + \sigma_b^2(Q_i(\theta)) + b^2(Q_i(\theta)) \right\},$$

where $\sigma_u^2(Q_i(\theta)) = \text{Var}_n(\tau_{0,i}(\theta)), \sigma_b^2(Q_i(\theta)) = \text{Var}_m(\tau_{1,i}(\theta))$ and $b(Q_i(\theta)) = \tau_{1,i}(\theta) - \tau_{0,1}(\theta)$. Let $\mathcal{F}_{s,i}(\theta) = \mathbb{E}_s(F_{s,i}(\theta))$ for $s \in \{0, 1\}$ and $\mathcal{F}_s(\theta) = \mathcal{F}_{s,L}(\theta) + \mathcal{F}_{s,R}(\theta)$, the population criterion is defined as $M_{of}(\theta) = \mathcal{F}_0(\theta) + \mathcal{F}_1(\theta)$. For the empirical criterion, we first define $\mathcal{F}_{n,i}(\theta) = \mathbb{E}_n(F_i^R(\theta))$ and $\mathcal{F}_{m,i}(\theta) = \mathbb{E}_m(F_i^R(\theta))$. Let $M_{N,of}(\theta) = \mathcal{F}_{n,L}(\theta) + \mathcal{F}_{n,R}(\theta)$ and $\mathcal{F}_m(\theta) = \mathcal{F}_{m,L}(\theta) + \mathcal{F}_{m,R}(\theta)$, the empirical criterion is the denoted as $M_{N,of}(\theta) = M_{N,of}(\theta) + \mathcal{F}_m(\theta)$. The population and empirical optimal splits are defined by

$$\theta_{of} = \arg\min_{\theta \in \Theta} M_{of}(\theta) \text{ and } \widehat{\theta}_f = \arg\min_{\theta \in \Theta} M_{N,of}(\theta).$$

We first have the following error decomposition:

$$M_{of}(\widehat{\theta}) - M_{of}(\theta_0) = M_{of}(\widehat{\theta}) - M_{of,\beta_n}(\widehat{\theta}_{of}) + M_{of}(\theta_0) - M_{of,\beta_n}(\theta_{of})$$
$$+ M_{of,\beta_n}(\widehat{\theta}_{of}) + M_{of,\beta_n}(\theta_{of}) - 2M_{N,of,\beta_n}(\widehat{\theta}_{of}) + 2M_{N,of,\beta_n}(\theta_{of})$$
$$+ 2M_{N,of,\beta_n}(\widehat{\theta}_{of}) - 2M_{N,of}(\widehat{\theta}_{of}) - 2M_{N,of,\beta n}(\theta_{of}) + 2M_{N,of}(\theta_{of})$$
$$+ 2M_{N,of}(\widehat{\theta}_{of}) - 2M_{N,of}(\theta_{of})$$
$$=: T_{1,n} + T_{2,n} + T_{3,n} + T_{4,n}.$$

By the definition of $\widehat{\theta}_{of}$, we have $T_{4,n} \leq 0$. In the following steps, we will bound $T_{1,n}, T_{2,n}$ and $T_{3,n}$, respectively. Following the same argument as for $S_{1,n}$, it can be obtained that $T_{1,n} \lesssim \frac{1}{n}$.

We now bound $T_{2,n}$ Let $\mathcal{G}_n = \{g : g(y,x) = \sqrt{1-w(\theta)}\tilde{y} - \tau)I(x \in \mathcal{Q}(\theta), \theta \in \Theta_n)\}$ and $\mathcal{H}_n = \{h : h(y,x) = \sqrt{w(\theta)}(\tilde{y} - \tau)I(x \in \mathcal{Q}(\theta), \theta \in \Theta_n)\}$, then via Lemma 2 we have

$$\mathcal{N}_1(\delta, \mathcal{G}_n, z_1^n) \leq (pn)^2 \left(\frac{c\beta_n}{\delta}\right)^4 \quad \text{and} \quad \mathcal{N}_1(\delta, \mathcal{H}_n, z_1^n) \leq (pn)^2 \left(\frac{c\beta_n}{\delta}\right)^4,$$

for any $\delta > 0$, where $z_1^n$ is any set $\{z_1, \cdots, z_n\} \in [0,1]^p \times \mathcal{Y}$ and $c > 0$ is a universal constant. It follows from Lemma 1 that for any $\alpha_1, \gamma_1 > 0$

$$\mathbb{P}_T \left\{ \exists \theta \in \Theta_n : |\mathcal{F}_{0,\beta_n}(\theta) - \mathcal{F}_{n,\beta_n}(\theta)| \geq \frac{1}{2}(\alpha_1 + \gamma_1 + M_{of,\beta_n}(\theta)) \right\}$$

$$\leq 28(pn)^2 \left(\frac{80c\beta_n^2}{\gamma_1}\right)^4 \exp\left(-\frac{\alpha_1 n}{1284\beta_n^4}\right)$$

$$\lesssim \exp\left(-\frac{\alpha_1 n}{\beta_n^4} + \log(pn) - \log(\gamma_1)\right).$$

Taking $\gamma_1 = 1/n$ and $\alpha_1 = (t + \log(pn))\beta_n^4/n$ implies that with probability at least $1 - C_4 e^{-t}$ for some universal constant $C_4 > 0$,

$$\forall \theta \in \Theta_n, |M_{of,\beta_n}(\theta) - 2\mathcal{F}_{n,\beta_n}(\theta)| \lesssim \frac{t + \log(pn)\log^4(n)}{n}. \tag{19}$$

Similary, for any $\alpha_2, \gamma_2 > 0$,

$$\mathbb{P}_O \left\{ \exists \theta \in \Theta_n : |\mathcal{F}_{1,\beta_n}(\theta) - \mathcal{F}_{m,\beta_n}(\theta)| \geq \frac{1}{2}(\alpha_1 + \gamma_1 + \mathcal{F}_{1,\beta_n}(\theta)) \right\}$$

$$\lesssim \exp\left(-\frac{\alpha_1 m}{\beta_n^4} + \log(pn) - \log(\gamma_1)\right).$$

Taking $\gamma_2 = 1/n$ and $\alpha_2 = (t + \log(pn))\beta_n^4/m$ implies that with probability at least $1 - C_5 e^{-t}$ for some universal constant $C_5 > 0$,

$$\forall \theta \in \Theta_n, |\mathcal{F}_{1,\beta_n}(\theta) - 2\mathcal{F}_{m,\beta_n}(\theta)| \lesssim \frac{t + \log(pn)\log^4(n)}{m}. \tag{20}$$

Combining (19) and (20) delivers that with probability at least $1 - 2C_1 e^{-t}$,

$$T_{2,n} \lesssim \frac{\log(pn)\log^4(n)}{m} + \frac{t + \log(pn)\log^4(n)}{n}, \tag{21}$$

for ant $t > 0$, since $\widehat{\theta}_{of}, \theta_f \in \Theta_n$ and $M_{of,\beta_n}(\theta) = \mathcal{F}_{0,\beta_n}(\theta) + \mathcal{F}_{1,\beta_n}(\theta)$ and $M_{N,of,\beta_n}(\theta) = \mathcal{F}_{n,\beta_n}(\theta) + \mathcal{F}_{m,\beta_n}(\theta)$ for any $\theta \in \Theta_n$.

Now we turn to $T_{3,n}$, the truncation error for the empirical loss. With the similar argument as in (16), we have with probability at least $1 - 4e^{-t}$, it holds that $T_{3,n} \lesssim (t+1)/n + (t+1)/m$ for any $t > 0$.

Combining the results for $T_{1,n}, T_{2,n}$ and $T_{3,n}$, we conclude that for any $t > 0$, with probability at least $1 - C_6 e^{-t}$,

$$M_{of}(\widehat{\theta}_{of}) - M_{of}(\theta_{of}) \lesssim \frac{t + \log(pn)\log^4(n)}{m} + \frac{t + \log(pn)\log^4(n)}{n}, \tag{22}$$

which completes our proof.

## D  Supporting lemmas

The following to lemmas are from Section 11.3 and Section 13.1 of Györfi et al. (2002), which are useful for our proofs.

**Lemma 1.** *(Deviation inequality of quadratic process). Suppose that $\mathcal{G}$ is a class of uniformly bounded functions $\mathcal{G} = \{g : \mathbb{R}^d \to \mathbb{R} \, \|g\|_\infty \leq M\}$. Let $\mathcal{F} = \{g^2 : g \in \mathcal{G}\}$. Then for any $n \geq 1$, it holds that*

$$\mathbb{P}\{\exists f \in \mathcal{F} : |\mathbb{E}\{f(z)\} - \mathbb{E}_n\{f(z)\}| \geq \varepsilon(\alpha + \gamma) + \mathbb{E}\{f(z)\}\}$$

$$\leq 28 \sup_{z_1^n} \mathcal{N}_1(\frac{\gamma\varepsilon}{20M}, \mathcal{G}, x_1^n) \exp\left(-\frac{\varepsilon^2(1-\varepsilon)\alpha n}{214(1+\varepsilon)M^4}\right),$$

*where $z_1^n = (z_1, \cdots, z_n) \in \mathbb{R}^d$, $\alpha, \gamma > 0$ and $0 < \varepsilon \leq 1/2$.*

**Lemma 2.** *(Covering number of piece-wise constant functions). Let $\Pi$ be the family of partitions of $[0,1]^p$. For any set $x_1^n = \{x_1, \cdots, x_n\} \subset [0,1]^p$, let $\Delta(\Pi)$ be the maximal number of partitions of $x_1^n$ induced by elements of $\Pi$. Let $M(\Pi)$ be the maximal number of sets contained in a partition $\mathcal{P} \in \Pi$. Denote the piece-wise constant functions on $[0,1]^p$ be $\mathcal{F}(\Pi)$ with $\|f\|_\infty \leq \beta_n$ for any $f \in \mathcal{F}(\Pi)$. Then using Lemma 13.31 and Theorem 9.4 of Györfi et al. (2002) we have*

$$\mathcal{N}_1(\delta, \mathcal{F}(\Pi), x_1^n) \leq \Delta_n(\Pi) \left( \frac{c_1 \beta_n}{\delta} \right)^{2M(\Pi)},$$

*for any $\delta > 0$, where $c_1 > 0$ is some univiersal constant. Specifically, in each partition for a node $\mathcal{C}_k$ of a tree, we have $M(\Pi) = 2$ and $\Delta_n(\Pi) \leq (pa_n)^2$, where $a_n$ is the sample size of $\mathcal{C}_k$.*

