# OpenReview forum: "FAST: a Fused and Accurate Shrinkage Tree for Heterogeneous Treatment Effects Estimation"
_NeurIPS.cc/2023/Conference — NeurIPS 2023 poster_

### Official Review · Reviewer_7DZZ · 2023-06-24

**Soundness:** 3 good
**Presentation:** 3 good
**Contribution:** 2 fair
**Rating:** 6
**Confidence:** 2

**Summary:**

This paper proposes a novel strategy for estimating the heterogeneous treatment effect called the Fused and Accurate Shrinkage Tree (FAST). The authors confirm the consistency of the proposed tree-based estimator and demonstrate the effectiveness of their criterion in reducing prediction error through theoretical analysis. The advantages of the proposed method over existing methods are demonstrated via simulations and real data analysis. As I am not very familiar with this field, it might be better to consider my opinion less.

**Strengths:**

1. This paper is technically sound.
2. The proposed method has better performance than the existing methods.

**Weaknesses:**

Imcomplete references: The tree-based method seems to not be a new method in this field. There might be some other references as follows.

Agarwal, Abhineet, et al. "Hierarchical Shrinkage: Improving the accuracy and interpretability of tree-based models." International Conference on Machine Learning. PMLR, 2022.

Nasseri, Keyan, et al. "Group Probability-Weighted Tree Sums for Interpretable Modeling of Heterogeneous Data." arXiv preprint arXiv:2205.15135 (2022).

**Questions:**

None

---

> ### Author Rebuttal · Authors · 2023-08-09
>
> Thanks for your instructive and detailed review comments. We are  encouraged by the overall positive responses from the comments and suggestions. Our point-to-point responses to your comments are itemized below.
>
> **Q1:**  Thank you for highlighting the potential absence of references. This suggestion greatly contributes to our effort to conduct a more comprehensive literature review.
>
>   After a careful reading of the two papers, we find them indeed closely related to our work. Specifically, Agarwal et al. (2022) also developed a shrinkage method to improve the performance of the tree estimator. One of the main differences is that our procedure shrinks the original estimator to a (potentially) biased estimator with small variability obtained from extra data sources, namely the observational data, while they propose to shrink the estimates at each leaf node to the sample means of its ancestors.
> Nasseri et al. (2022), on the other hand, generalized the tree-based methods to address the challenge of heterogeneous data from diverse data sources. It is worthwhile to note that both methods can be readily applied together with our shrinkage strategy, as long as a sufficient amount of (potentially biased) observational data is available.
>
> We would discuss these aspects in the revised version of the paper if given a chance.

---

> > ### Comment · Reviewer_7DZZ · 2023-08-10
> >
> > Thank you for your response. This address my concern. Since I am not expert in this field, I will keep my score.

---

### Official Review · Reviewer_jsqm · 2023-06-30

**Soundness:** 2 fair
**Presentation:** 2 fair
**Contribution:** 2 fair
**Rating:** 3
**Confidence:** 3

**Summary:**

The paper deals with the problem of estimating the heterogeneous treatment effects with multiple data sources. In particular, the paper aims to utilize the information from the observational data to better estimate the causal effects in the trial data. Inspired by the shrinkage estimation, a weighting scheme is developed to balance the unbiased estimator based on trial data and the potentially biased estimator based on observational data. Specifically, a tree-based algorithm with new split criterion is proposed based on above motivations. Some theoretical results about the causal effect estimation is derived. Finally, the author provides simulations and real data analysis to demonstrate the performance of the proposed method.


**Strengths:**

The papers deal with an interesting problem in practice, i.e., data fusion. In particular, we may have multiple data sources. However, some sources has limited observations with unbiased causal effects and other sources have sufficient observations with biased causal effects. The paper utilizes the tree-based algorithm with a new splitting criterion to tackle this issue. In addition, some theoretical analysis are provided to prove the advantages of the method.

**Weaknesses:**

1. Although the paper considers an important problem in practice, the reason why the method chooses tree-based algorithm is not convincing. In particular, other ML methods can also achieve data fusion. The advantages of using tree-based algorithm over other methods are confusing and are not clearly discussed.
2. In the introduction and simulation studies, the author also mentions many other methods that deal with the data fusion problem. However, the reason why the proposed tree-based method can perform better than other methods are not interpreted.
3. Although the paper provides the theoretical analysis for the causal effect estimation, the interpretation for the theorem are not convincing. For example, many other methods also have theoretical guarantee for the causal effect estimation. Which part in the theoretical analysis can illustrate the advantages of data fusion and the tree-based algorithm?
4. In the general picture, the idea of data fusion is very similar to that of transfer learning, i.e., we want to transfer the information from the observational studies to help the estimation of causal effects in trial data. However, the paper does not mention any related literature in transfer learning. In particular, what is the advantages and differences of the method compared with transfer learning? A more comprehensive literature review is encouraged.

**Questions:**

1. When the bias of the causal effect estimation in observation studies is large, will the data fusion cause any harm to the estimation of the causal effects? Can it be interpreted as a trade-off between the bias and variance?
2. In section 2.1, the author mentions $\hat{e}(X,1)$ is unbiased while $\hat{e}(X,0)$ is biased. Why is that? Didn't they come from the same data source with $\hat{e}(X,1) + \hat{e}(X,0) = 1$?
3. One of the main questions is that, why using tree-based methods for data fusion?
4. In real data analysis, the author mentions that the true causal effect is unknown and hence using the estimation of the generalized random forest as the ground truth. This is confusing. If the estimation using random forest is not accurate, then the results in the real data analysis is not reliable and convincing. How do the author correct this potential bias?

**Limitations:**

Please see in Weaknesses.

---

> ### Author Rebuttal · Authors · 2023-08-09
>
> Thanks for your careful review. Our point-to-point responses are as follows and we would add the additional discussions in the revised manuscript if given a chance.
>
> **W1&W2&W3&Q3**:
> Thanks for raising the issues concerning tree-based methods. As mentioned in the abstract of the paper, one of the key contributions of the paper is the establishment of a new data fusion framework called shrinkage estimation. The second is the implementation of a tree-based method by developing an adaptive fusion criterion in Section 3.2. And we chose tree-based methods to show the core idea of the shrinkage estimation framework for two main reasons:
> - the advantages of tree models on tabular data, e.g., robustness to features and model interpretability;
> - The core of this fusion framework is to estimate the variance and bias, which further facilitates the estimation of $w^*$. Although the estimator mentioned here can be any ML model e.g., neural network (NN), many of them cannot conveniently calculate the estimator’s variance. For the tree model,  the variance $σ_u^2$ of the trial estimator can be easily estimated based on the observations that fall into the corresponding tree leaf.
>
> Besides, the rfFAST does not require any parametric models on the data generation mechanism, which ensures its robustness. In comparison, existing methods in the literature usually rely on parametric assumptions. This may explain the better performance of the rfFAST over other existing methods.
>
> Finally, for the theoretical analysis, as stated in Line 210 of the manuscript, "we formally establish the benefits of the proposed split criterion (9) compared with the conventional criterion (7)",  which can be specifically listed as follows:
>
> - Theorem 1 shows that our fused estimator with the novel proposed criteria enjoys a uniform MSE reduction property compared to the conventional method using only the trial data.
> - It further takes finite sample variations into account and establishes non-asymptotic bounds for the excess risks of the empirical solutions of the conventional and proposed tree split criteria, respectively, ensuring that the MSEs of the estimated tree are close to those of optimal solutions under the population level.
>
> **W4** Thanks for your suggestion regarding a comparison between transfer learning (abbr TL) and causal data fusion. Intuitively, if we simply consider observational data as the source domain and trial data as the target domain, the trial and observational data fusion scenario can be regarded as a domain adaption problem.
> However, there are a few significant differences between these two concepts.
> - Our purpose is to identify cause-and-effect relationships between different variables, while traditional TL focuses on the predictive task. However, The main motivation of TL is to leverage the labeled data of the target domain to improve the performance of the prediction task (Pan and Yang, 2010; Weiss et al., 2020).
> - TL aims to learn the shared knowledge of domains while distinguishing the specific knowledge. In comparison, the prerequisite for causal data fusion is that both the trial data and the observational data share the same causal effect function, as stated in Assumption 1. Besides,  in general, the ground truth is inaccessible for causal data fusion.
>
> **Q1** Indeed, the core idea underlying the shrinkage method is closely related to the classical "bias-variance" trade-off in ML. Recalling some notations:
> $$σ_u^2=Var(\hat{τ}_u^2),b=\mathbb{E}(\hat{τ}_b-\hat{τ}_u), w^*=\frac{σ_u^2}{σ_u^2+b^2},$$
>  and the MSE of  the optimal fused estimator $\hat{τ}\_{w^*}=w^*\hat{τ}_b+(1-w^*)\hat{τ}_u$ can be expressed as $MSE(\hat{τ}\_{w^*})=(1-w^*)σ_u^2$. From the expression of $w^*$, it becomes apparent that the optimal weight seeks an equilibrium between the squared bias $b^2$ of the observational estimator and the variance $σ_u^2$ of the trial estimator.   Also, it is worth mentioning that this interplay between bias and variance is easily achieved through our shrinkage method for real applications, as outlined in the paper.
>
> **Q2** The symbol $S$ (the second parameter of the function $\hat{e}$) serves as an indicator, and $S=1$ means that the individual is sampled from the trial population otherwise from the observational population. And $e(X,U,S)=P(D=1|X,U,S)$ means the conditional probability of being selected to the treatment group of an individual.
>
> First, as outlined in Lines 72 to 74, "In practice, Due to $U$ being unknown, we usually use $\hat{e}(X,S)$ to estimate $e(X,U,S)$. In addition, $\hat{e}(X,1)$ is unbiased for the randomization of trial data, but $\hat{e}(X,0)$ is biased because the unmeasured confounder $U$ is related to the assignment of treatment $D$".
>
> In general, $\hat{e}(X,1)+\hat{e}(X,0)$ means $P(D=1|X,S=1)+P(D=1|X,S=0)$ which doesn't equal to $1$.
>
> **Q4**  We first trained a generalized random forest using the full STAR dataset. We then regarded the resulting estimator as a surrogate of the underlying ground truth. The reasons are listed below.
> - It is worth mentioning that a generalized random forest (GRF) estimator is consistent under mild conditions (Athey et al., 2019), meaning that as the sample size increases, it converges to the true causal effect. In our case, since the full dataset is collected from a randomized controlled experiment, then the estimation error of the GRF estimator is expected to be quite small given its sample size.
> - Unlike other fields of machine learning, in causal inference, the ground-truth causal effect is typically inaccessible.  From this perspective, a certain degree of approximation is required. And similar approaches have been used in the literature (Kallus et al., 2018).
>
> **Some Refs:**
> - Pan, S. and Yang, Q. (2010)  A survey on transfer learning. IEEE Transactions on knowledge and data
> engineering, 22(10):1345–1359.
> - Weiss, K., Khoshgoftaar, T. M. and Wang, D. (2016). A survey of transfer learning. Journal of Big Data, 3(1), 1-40.

---

> > ### Comment · Reviewer_jsqm · 2023-08-18
> >
> > Thanks for the authors providing the reply to my questions. For our concerns and questions about why using tree-based models in data fusion, although the authors list some advantages for the tree-based methods, most of them are from general perspective. It is still lack of sufficient support about why using specific tree methods in data fusion. In addition, for the concern about the true value of causal effects, it is still questionable that whether the estimation from GRF is correct. Although GRF has theoretical guarantee, we cannot make sure it happens in real data analysis. So, the comparison results may be lack of strong conclusions.

---

> > > ### Author Response · Authors · 2023-08-19
> > >
> > > Thanks for your comments.
> > >
> > > **1.   For your concern regarding the use of tree-based methods:**
> > >
> > > Besides the general benefits of tree-based methods as outlined in the previous reply,   we pointed out in second point of the first answer, that the greedy and local averaging nature of  tree-based algorithms makes it extremely simple to implement the shrinkage estimation framework, as the estimator of the key gradient $w^*$ can now be easily obtained.
> > > This computational and notational simplicity help the readers better understand the core idea of the shrinkage concept.
> > >
> > >
> > > **2.  For your concern regarding the validity of the full-sample GRF estimator in the real-data analysis:**
> > >
> > > - Firstly, its correctness can  be empirically verified via the numerical results of the real-data analysis presented in Figure 3 of the manuscript: Except the SF (simple fusion) estimator, which is not consistent and performed badly in the simulations, the mean square differences between the remaning estimators and the full-sample GRF estimator did exhibit a downward trend as the sample size of the unbiased data increased. These trends couldn’t exist if the full-sample GRF estimator was wrong.
> > > - Besides, as metioned in the comments and reply, the GRF estimator is consistent for unconfounded data.  That is to say,   its estimation error vanishes as the training sample increases. Thus, given the large training sample size of the full data,  the estimator should be accurate from a theoretical perspective.
> > > -  Lastly, since the ground-truth of causal effect is inaccessible in real applications, a certain degree of approximation is necessary. And some recent existing works applied similar approaches to construct surrogates for the inaccessible ground-truth (Kallus et al., 2018; Wu et al., 2022).
> > >
> > > We would add these discussions in the revised version if given a chance. And we would kindly ask you for more specific questions concerning the proposed method itself, so that we could continue our revision, and we are willing to provide further explanations. Your continued guidance is greatly appreciated.

---

### Official Review · Reviewer_7Nsn · 2023-07-06

**Soundness:** 4 excellent
**Presentation:** 4 excellent
**Contribution:** 3 good
**Rating:** 7
**Confidence:** 4

**Summary:**

The authors propose a novel shrinkage method that fuses an unbiased estimator with a biased estimator. This method effectively reduces the MSE of the unbiased estimator. The approach offers a practical and straightforward implementation specifically tailored for estimating heterogeneous treatment effects. The authors extend the conventional node split criterion to align with the fused estimator and penalizes the use of observational data with substantial confounding bias. The authors also provide a theoretical analysis that explains the advantages of the modified splitting criterion.

**Strengths:**

- The application of the weighting strategy from shrinkage estimation to fusing unbiased and biased estimators in order to reduce the MSE of the unbiased estimator is a great idea.
- The modification of the node-splitting criterion that aligns with the fused estimator is an excellent enhancement to the methodology.
- The paper is well-organized and thanks to the authors's thoughtful and consistent notation, the methodology is easy to follow.

**Weaknesses:**

I'm concerned with the omission of $\sigma^2_b$ in practice. If we only look at the weight $w$, it make sense if $\sigma^2_b$ is small comparing to $\sigma^2_u$. However, in the tree building process, we need working estimates of the MSE of fused estimator, $\frac{(\sigma^2_b+b^2)\sigma^2_u}{\sigma^2_b+b^2+\sigma^2_u}$, and I don't think omitting $\sigma^2_b$ is justified by the same reason anymore.

**Questions:**

The sample sizes considered in the experiment are always large. If a sample from RCT of size 100 or 200 is avaliable, the baseline models would do the job. I'm more curious in the senario when we have a relatively small RCT sample and that we do need more data from obervational studies to enhance the estimation.

**Limitations:**

No other limitation.

---

> ### Author Rebuttal · Authors · 2023-08-09
>
> Thanks for your instructive review comments. We are greatly encouraged by the overall positive responses from the comments and suggestions. Our point-to-point responses to your comments are itemized below and we would add those discussions in a revised manuscript.
>
> **W1:** Thanks for pointing out the issue. The MSE of the fused estimator can be equivalently expressed as
> \begin{equation}
> \frac{(\sigma_b^2 + b^2)\sigma_u^2}{\sigma_b^2 + b^2 + \sigma_u^2} = (1 -\frac{\sigma_u^2}{\sigma_b^2 + b^2 + \sigma_u^2})\sigma_u^2 = (1-w^*)\sigma_u^2,\nonumber
> \end{equation}
> so working estimates of the MSE of the fused estimator amount to estimating the optimal weights $w^*$ and the variance of the trial estimator $\sigma_u^2$. Thus, we would think that the same reasoning can be applied to both the estimation of $w^*$ alone and the MSE of the fused estimator. This also can be considered to be a significant benefit of applying the proposed shrinkage method: in real applications, once the optimal $w^*$ is estimated, one can immediately obtain an estimate of the MSE of the corresponding fused estimator. Furthermore, the relative performance improvement of the fused estimator over the original trial estimator is readily at hand -- it is expected to be exactly $w^*$.
>
> **Q1:**      Thanks for raising this issue. Indeed, as mentioned in your comment, one motivation for developing data fusion methods is to tackle the inadequacy of the RCT sample in real applications. Following your suggestions, we have enriched the original Table $1$ (see Table 1 in the global response) by adding a scenario when the trial sample size $n$ is reduced to $50$. We added the subscripts ''NF'' and ''SF'' to represent the trial-data-only and the simple fusion method (the two simple fusion methods were also included for completeness), which makes it easier to compare the performances of all the methods presented in the Table.
>     When $n=50$, the $\mathrm{rfFAST}$ still dominantly outperformed other data fusion methods and led to at least a 20 percent improvement in performance compared with its no fusion counterpart $\mathrm{GRF}_{NF}$.

---

### Official Review · Reviewer_tnj1 · 2023-07-07

**Soundness:** 3 good
**Presentation:** 3 good
**Contribution:** 4 excellent
**Rating:** 7
**Confidence:** 3

**Summary:**

The paper introduces a Fused and Accurate Shrinkage Tree (FAST) algorithm for heterogenous treatment effect estimation given trial and observational data. The FAST algorithm introduces (i) a shrinkage based approach that combines trial and observational data for MSE reduction in treatment effect estimation, and (ii) a split criteria which down-weights observational data with high confounding bias. Further, the paper provides theoretical analysis demonstrating the benefits of the proposed split criteria. Experimental results on synthetic and real-world data demonstrate that the proposed approach outperforms baselines per metric MSE.

**Strengths:**

- The paper is well written and easy to follow. The reviewer enjoyed reading this paper.
- The proposed FAST algorithm is well-justified and the theoretical analysis might be of interest to some readers.
- The paper tackles an important problem (fusing small RCT with large readily available observational data ) with many applications.
- The algorithm seems simple and easy to implement

**Weaknesses:**

-  Eqn. 5:  The paper seems to have glazed over the rationale for dropping $\sigma_b$ in the shrinkage estimator. It's unclear in what scenarios, e.g., how large the observational sample size must be for $\sigma_b$  to become negligible.

**Questions:**

**Experiments**
- Table 1: Could you include results for the NF estimator (using trial data only) and SF estimator (both trial and observation data)
- Figure 3: Could you comment on why NF is not monotonically decreasing with sample size? Shouldn't we expect the rFAST to NF estimators to cross when $n$ is large enough?

**There are some limitations with tree-based methods that the paper does not adequately address:**
- How sensitive is the FAST algorithm to the dimension $p$ and heterogeneity of the covariates? The experiments set $p=5$, which is unrealistic in real-world scenarios.
- What is the complexity, e.g., training time of the rFAST algorithm compared to baselines, and how easy is it to scale given high-dimensional heterogeneous covariates?


Minor:
- Figure 1: Could you add more details summarising the plots in the caption


**Limitations:**

The limitation discussion is inadequate. I encourage the authors to add a paragraph discussing the limitations.

---

> ### Author Rebuttal · Authors · 2023-08-09
>
> Thanks for your inspiring review comments. We are greatly encouraged by the overall positive responses from the comments. Our point-to-point responses are itemized below and the references are listed in the end. We would add the discussions and numerical experiments to the revised version if given a chance.
>
> **W1:** Thanks for pointing out the missing details in dropping the $\sigma_b$ term. In fact, as pointed out in Section 3.1, both  $\hat{\tau}_u$ and  $\hat{\tau}_b$ are approximately sample means, so the variances of both estimators should admit:
> $$
> \sigma_u^2 = Var(\hat{\tau}_u) = \frac{C_u}{n} \ \hbox { and } \ \sigma_b^2 = Var(\hat{\tau}_b) = \frac{C_b}{m},
> $$
> where $C_u, C_b$ are positive constants, and $n$ and $m$ are the  sample sizes. Thus, if we denote the dropping-$\sigma_b$ version of $w^*$ as $\tilde{w}^*$, then
> $$
> |1 - \frac{\tilde{w}^*}{w^*}| = |1 -\frac{b^2 +\sigma_u^2}{b^2 + \sigma_b^2 + \sigma_u^2}| = |\frac{1}{1 + m(\frac{b^2 + \frac{C_u}{n}}{C_b})}| = O(\frac{1}{m(b^2 + n^{-1})}).
> $$
> Thus for a given tolerance $\epsilon > 0$,  the condition for the obs sample size $ m > \frac{1}{\epsilon (b^2 + n^{-1})}$ suffices for $\sigma_b$ to become negligible.
>
>
> **Q1:** We have added the results of the SF estimators (the $HT\_{SF}$ and the $GRF\_{SF}$) in Table 1 of the additional PDF. Besides, both the $HT$ and $GRF$ estimators in the original Table 1 are in fact  NF estimators. We added the subscript ''NF'' to address this aspect. From the new Table 1, the estimator $GRF\_{SF}$ performed inferior to the $GRF\_{NF}$ in all scenarios. And the single-estimator method $HT_{SF}$ exhibited smaller MSEs compared to the $HT_{NF}$ only when $\beta$ is small. Both SF estimators performed worse than the  $rfFAST$ estimator.
>
> Second, unlike other fields of machine learning,  the actual causal effect is typically inaccessible in causal inference. So we used the generalized random forest estimator trained on the full dataset as a surrogate for the underlying inaccessible ground truth. And similar approaches have been used in literature (Kallus et al., 2018). Thus, the nonmonotonicity is largely due to the intrinsic randomness of the data. Besides, the $\mathrm{NF}$ estimator in the Figure does show a downward trend, which is consistent with our simulation results if we check the column of the $\mathrm{GRF}_{NF}$ estimator of Table 1 of the additional PDF. Besides, as the theoretical MSE of $\mathrm{rfFAST}$ estimator is $(1-w^*)\sigma_u^2$ and that of the $NF$ estimator is $\sigma_u^2$, the proposed fused estimator $\mathrm{rfFAST}$ should always perform better than its no fusion counterpart $NF$.
>
> **Q2:**.  First, since the $FAST$ is essentially a tree-based method, its sensitivity to the dimension $p$ and heterogeneity of the covariates largely resembles that of the classical decision tree. And one of the appealing advantages of tree-based methods compared with other nonparametric methods is that they can be flexibly adaptive to high-dimensional and complex features, as revealed both empirically (Archer et al., 2008) and theoretically (Chi et al., 2022).  To see this, we conducted an additional experiment  presented in Table 3 in the additional PDF. We did not include $SF$ methods to save space. And $rfFAST$ estimator was quite robust against increasing  $p$.
>
> Second, we admit that our current code is only used to implement the method prototype in Python, and we have not done much optimization. And the actual running time of the algorithm depends on many factors, such as the different implementation languages (GRF is based on C++), histogram preprocessing acceleration like LightGBM, and parallelization. Therefore, here we analyze the theoretical time complexity.  For data with $n+m$ samples and $p$-dimensional features, the time complexity of building a tree by our method is $O(p\cdot (m+n)\cdot\log((m+n))+p\cdot n\cdot\log(n))$, where the extra overhead compared to the decision tree is mainly due to calculating $w^*$, that is, the second term. But according to equation (2), $w^*$ has a closed-form solution, and the computational complexity is $O(pn\log n)$. Given that $n\ll m$, the overall computational complexity $O(p\cdot (m+n)\cdot\log((m+n))$ is approximately the same order as that of the traditional decision tree. Therefore, there is no significant difference between the two methods.
>
> **Q3**. We re-wrote the caption of Figure 1:  ''The probability density functions (pdfs) of the unbiased estimator (pink) and the biased estimator (blue) in the left panel and the pdf of the shrinkage estimator under the optimal weight $w^*$ (green) in the right panel. The vertical dashed line represents the true parameter value $\theta^* =0$. '
>
> **Q4**. We added a paragraph discussing the limitations: Our work also has several limitations. First, as mentioned above, our method currently can not provide a confidence interval for the fused estimator.  Second,   we opt for the mean square error (MSE) criterion to attain an optimal balance between the variance of the trial estimator and the bias of the observational estimator. This choice is motivated by the existence of a readily obtainable closed-form expression for the optimal weight $w^*$ under this criterion, thereby enhancing its interpretability. The consideration of alternative criteria for optimization remains untouched.
>
>
> **Refs:**
> - Archer, K. J. and Kimes, R. V. (2008). Empirical characterization of random forest variable importance
> measures. Computational statistics & data analysis, 52(4):2249–2260.
> - Chi, C.-M., Vossler, P., Fan, Y., and Lv, J. (2022). Asymptotic properties of high-dimensional random
> forests.The Annals of Statistics, 50(6):3415–3438
> - Kallus, N., Puli, A. M., and Shalit, U. (2018). Removing hidden confounding by experimental grounding. In Bengio, S., Wallach, H., Larochelle, H., Grauman, K., Cesa-Bianchi, N., and Garnett, R., editors, Advances in Neural Information Processing Systems, volume 31. Curran Associates, Inc.

---

> > ### Comment · Reviewer_tnj1 · 2023-08-17
> > **Official Comment by Reviewer tnj1**
> >
> > Thanks for addressing all my comments.

---

### Official Review · Reviewer_YpW7 · 2023-08-01

**Soundness:** 3 good
**Presentation:** 3 good
**Contribution:** 3 good
**Rating:** 6
**Confidence:** 3

**Summary:**

This paper considers the problem of estimating (heterogeneous) treatment effects via both interventional and observational data. The authors proposed a new estimation, namely the Fused and Accurate Shrinkage Tree (FAST), which optimally weights the interventional and observational estimator, and combines with a new spilt criteria for tree-based heterogeneous treatment effect estimation. The authors further conducted experiments to compare FAST against existing methods.

**Strengths:**

- Apart from a few typos, the paper is well written and ideas are presented in a rigorous but clear also way.
- The idea of applying shrinkage method to combine interventional and observational data for better estimator is novel and could be a nice addition to the literature.
- Note that I have not gone through all the proofs in the appendix, the mathematical correctness might need further input from other reviewers.

**Weaknesses:**



Generally I like this paper, but there are still a few weaknesses.

- The main issue with shrinkage method is interpretability: we need to understand better how the variance-bias trade-off behaves in different regimes, especially the authors takes a more analytic way to first estimate the required quantities, then solve the optimal weights. More specifically for example, it would be beneficial to at least see different how trial mechanisms affects the estimation. For example, in a more realistic setting, one may consider non-randomized trials rather than RCT, in which treatments are assigned by a *known* true model. By adjusting the parameters of such true assignment model, the estimator variance for the trial population HTE estimator can be controlled (even with fixed N). Then the performance of FAST can be evaluated against different variance regimes of trial HTE estimator, which will help us understand the sweat spot of the method.

- Regarding experimental settings. It is indeed quite standard for these type of papers to have 1 or 2 synthetic experiments and 1 real data experiment. However, in the case of this paper I found the simulation setting is a bit weak. It would be great to perform experiments on multiple data generating mechanisms with randomly sampled parameters and coefficients, allowing us to evaluate the marginal performance of the method. Otherwise the authors at most demonstrated the capability of the method on only one single data generation mechanism (which arguably is much easier to hack/cherry-pick).


- The other potential room for improvement is the baseline. I understand that the paper mainly only compares to data fusion methods. However, due to the variance-bias trade-off of the shrinkage method, it would be natural to also expect some comparisons to variance reduction methods for trial estimators as well.

**Questions:**

See above.

**Limitations:**

The authors has somewhat discussed the limitations of the method.

---

> ### Author Rebuttal · Authors · 2023-08-09
>
> Thanks for your insightful and comprehensive review. We are encouraged by the overall positive responses from the comments and suggestions. Our point-to-point responses to your comments are itemized below and we would add the discussions and numerical experiments in the revised manuscript if given a chance.
>
> **W1:** Our apologies for any confusion that might have arisen due to the unclear illustration. It is worthwhile to first  note that for any regular unbiased estimator $\hat{\tau}_u$ based on the trial data and (potentially) biased estimator $\hat{\tau}_b$ based on the observational data, the variance of both estimators should admit the following expressions:
> \begin{equation}
> \sigma_u^2 = \mathrm{Var}(\hat{\tau}_u) =  \frac{C_u}{n} \ \hbox { and } \ \sigma_b^2 = \mathrm{Var}(\hat{\tau}_b) = \frac{C_b}{m},\nonumber
> \end{equation}
> where $C_u, C_b$ are positive constants, and $n$ and $m$ are the sample sizes of the trial data and the observational data, respectively.
>
> Indeed, as mentioned in your comment, by applying various methods including adjusting the trial mechanisms (either randomized or non-randomized), one is able to control the variance for the trial population estimator $\sigma_u^2$ only through the constant factor $C_u$ without changing the order $n^{-1}$. But the implementation of our shrinkage method is independent of the trial mechanism. To see this, for a given trial mechanism and estimation method (mapping to a $C_u$), one can readily apply the shrinkage method to obtain the optimal fused estimator $\hat{\tau}\_{w^*}$, whose mean square error (MSE) is $(1 - w^*)\sigma_u^2$. Now,  the relative improvement of our proposed estimator $\hat{\tau}\_{w^*}$ over the original trial estimator in terms of MSE is as follows:
> \begin{equation}
> 1 - \frac{\mathrm{MSE}(\hat{\tau}_{w^*})}{\mathrm{MSE}(\hat{\tau}_u)} = w^*, \hbox{ where } w^* = \frac{\frac{C_u}{n}}{b^2 + \frac{C_u}{n} + \frac{C_b}{m}}.\nonumber
> \end{equation}
> From the above equation, one can find that $w^*$ is not only an optimal weight facilitating the shrinkage estimation, but it itself also characterizes the extent of improvement in terms of MSE one can expect via incorporating the observational data. Thus, in real-world applications, one is able to get an estimate of the improvement of the procedure once $w^*$ is estimated.
>
> **W2:**  In response to this suggestion, we designed the following two data-generating processes (DGPs). In both DGPs,  we generated the pre-treatment covariates $X = (X_1, X_2, \cdots, X_p)^T$ from $\mathrm{Uniform}[-1,1]^p$, $U$ from $N(0,1)$, $D|(X,U, S=1) \sim Ber(0.5) $ and  $\epsilon(d) \sim N(0,1)$.
>
> -  (I)
> $$    Y(d) = d\tau(X,\beta_{\tau}) + X^T\beta_{\ell} + \beta_U U + \epsilon(d), \beta_{\tau} = (\beta_{\tau,1},\beta_{\tau,2},\cdots,\beta_{\tau,5})^T \sim N(1_{5\times 1}, 0.5^2I_5) , \beta_{\ell} = (\beta_{\ell,1},\beta_{\ell,2},\cdots,\beta_{\ell,p}) \sim N(1.5_{p\times 1}, 0.5^2I_5),\beta_U \sim N(1.5,0.5^2),
>  $$
> $$
>  \tau(X,\beta_{\tau}) = \beta_{\tau,1} + \beta_{\tau,2}X_1 + \beta_{\tau,3}X_1^2 + \beta_{\tau,4}X_2 + \beta_{\tau,5}X_2^2, D|(X,U,S=0) \sim Ber(1/(1+\exp(\beta U + \beta_oX_1))), \beta_{o} \sim N(1.5, 0.5^2)
>   $$
> - (II)
> $$Y(d) = d\tau(X, \beta_\tau) + \beta_{\ell,1}\sin{X_1} + \beta_{\ell, 2}\cos{X_2} + \beta U + \epsilon(d), \beta_{\tau} \sim Uniform(0.5,1.5)^p, \beta_{\ell} \sim N(1_{2\times 1}, 0.5^2I_2), $$
> $$\tau(X,\beta_{\tau}) = 1 + \sum_{s=1}^p\beta_{\tau,s}\left(X_s+X_s^2\right), D|(X,U,S=0) \sim Ber(1/ (1+ \exp(\beta U + \beta_{o,1}X_1 + \beta_{o,2}X_2))), \beta_{o,i}\sim N(1.5,0.5^2).$$
>
> It is noted that in (II), the factor $\beta$  appears in both  $Y(d)$  and $D|X, U, S=0$. We repeated each DGP 100 times and the results are presented in Table 2 in the additional PDF. In both DGPs, the proposed $\mathrm{rfFAST}$ estimator performed the best in most scenarios. Furthermore, its advantage over other data fusion methods was more obvious when the trial sample size was small ($n=100$),  which is exactly when data fusion is necessary to improve the causal effect estimation in real-world applications.
>
>
> **W3**.  Thanks for your understanding of our focus on comparing with data fusion methods. Following your suggestion, we have done another round of literature review regarding variance reduction methods for trial estimators:
>
> - Li, F., Morgan, K.L. and Zaslavsky, A.M. (2018) Balancing covariates
> via propensity score weighting. Journal of the American Statistical
> Association, 113(521), 390–400.
>  -  Sturmer T. et al., (2021). Propensity score weighting and trimming strategies for reducing variance and bias
> of treatment effect estimates: a simulation study. American journal of epidemiology, 190(8):1659–1670.
> - Liao, J. and Rohde. C. (2022). Variance reduction in the inverse probability weighted estimators for the
> average treatment effect using the propensity score. Biometrics, 78(2):660–667.
>
>
> We believe that these references will definitely enrich our literature review and make it more comprehensive. However, two points are also worth mentioning: First,  while indeed the $\mathrm{HT}$ estimator based on the inverse probability weighting (IPW) is sometimes considered to be unstable,  the $\mathrm{GRF}$ estimator namely the generalized random forest estimator already have certain satisfying statistical properties verified both theoretically and empirically (Athey et al., 2019), where implicit variance reduction techniques have been applied. The other one is closely related to the core feature of our shrinkage method. That is, the proposed framework is essentially built upon the trial and observational estimators, namely the implementation of the procedure is independent of the construction of the trial estimator. In real applications, the practitioners can first construct a variance-reduced trial estimator designed for their particular scenarios, then estimate the optimal weight $w^*$ to facilitate the shrinkage estimation.

---

### Author Rebuttal · Authors · 2023-08-09

We would like to express our sincere thanks to all the reviewers for your insightful and constructive review comments and we are greatly encouraged by the overall positive responses. Based on your valuable suggestions, we have carefully done a round of revision of the manuscript. While detailed responses to individual points can be found in the subsequent rebuttals, we also seize this moment to summarize the progress achieved in improving this paper over the past week.

-  We improved the illustration of the core idea of the shrinkage estimation framework.
-  A more detailed explanation of the expression of the optimal weight was given and missing details of the derivations due to page limit were presented.
-  The algorithm complexity of the FAST estimator was analyzed.
-  The advantages of the proposed method over existing methods were interpreted and limitations were discussed.
-  Three additional numerical experiments were conducted (shown in the additional PDF file) to better validate the effectiveness of the proposed method.

We would add all the revisions to the manuscript if given a chance. Thanks again for all your great efforts and contributions in jointly improving this work.

---

### Comment · Area_Chair_L55j · 2023-08-18
**Reviewers, please respond to author's rebuttal.**

As a minimum, please acknowledge that you have read the rebuttal and whether it helps to change your rating, as the authors have tried to respond to your comments in the review. Thank you.

---

### Author Response · Authors · 2023-08-21
**Thanks for the careful review of the paper**

Dear chairs and reviewers,

First and foremost, we would like to express our sincere gratitude to you and the reviewers for your hard work and efforts in evaluating our submission. And we would also like to thank all reviewers for their constructive and helpful comments. With the help of your comments and suggestions, we have thoroughly revised our manuscript. Besides, we have also provided an explaination for each comments in a point-by-point style.

Specifically, in order to thoroughly enhance the readability and clarity of our paper, we have followed the suggestions of the reviewers and made a concerted effort to elucidate the ideas behind our approach, as well as to introduce the relevant background and concepts in a more detailed and clear manner. We anticipate that these enhancements would help to eliminate ambiguity and facilitate a better understanding. In addition, as requested by the reviewers, we have added more simulation experiments to support our claims; these simulated results further enhance our confidence in the effectiveness and robustness of the proposed approach - the Fused and Accurate Shrinkage Tree (FAST).

Limited by the issues related to cost, time, and ethics, large-scale experimental data (also known as randomized control trials) is unavailable. Obstructed by the possibility of confounding bias, the heterogeneous treatment effect on observational data usually is non-identifiable. Our work leverages the shrinkage estimation in statistic to develop an optimal weighting scheme and a corresponding data-driven estimator that balances the unbiased estimator based on trial data with the potentially biased estimator based on observational data. Considering advantage of tree-based techniques e.g, robustness, interpretability, and especially friendliness to implement shrinkage estimation framework, we implement a novel tree-based fusion method. Moreover, we confirm the consistency of our proposed tree-based estimator and demonstrate the validity and utility through theoretical analysis and numerical studies. We are encouraged that reviewers find our paper targets at an interesting and important problem in causal inference (Reviewers tnj1 and YpW7), proposes a novel fusion framework and estimation methodology (Reviewers 7Nsn and YpW7). We are also gald to receive feedback from the reviewers confirming that our detailed responses effectively addressed the relevant questions and doubts.

Yours Sincerely,

Authors

---

### Decision · Program_Chairs · 2023-09-21

**Decision:**

Accept (poster)

**Comment:**

The reviewers appraised the paper is well written and easy to follow, that the proposed algorithm is well-justified and the theoretical analysis is interesting, that the underlying problem is important, and the idea of applying shrinkage method to combine interventional and observational data for better estimator is novel.

There is one low rating but in my opinion it is somewhat offset by other high ratings. Overall, I agree with the over majority of the reviewers and support acceptance of this paper.